# ADMETA: A NOVEL DOUBLE EXPONENTIAL MOVING AVERAGE TO ADAPTIVE AND NON-ADAPTIVE MOMENTUM OPTIMIZERS WITH BIDIRECTIONAL LOOKING

## ABSTRACT

Optimizer is an essential component for the success of deep learning, which guides the neural network to update the parameters according to the loss on the training set. SGD and Adam are two classical and effective optimizers on which researchers have proposed many variants, such as SGDM and RAdam. In this paper, we innovatively combine the backward-looking and forward-looking aspects of the optimizer algorithm and propose a novel ADMETA (**A D**ouble exponential **M**oving averag**E T**o **A**daptive and non-adaptive momentum) optimizer framework. For backward-looking part, we propose a DEMA variant scheme, which is motivated by a metric in the stock market, to replace the common exponential moving average scheme. While in the forward-looking part, we present a dynamic lookahead strategy which asymptotically approaching a set value, maintaining its speed at early stage and high convergence performance at final stage. Based on this idea, we provide two optimizer implementations, ADMETAR and ADMETAS, the former based on RAdam and the latter based on SGDM. Through extensive experiments on diverse tasks, we find that the proposed ADMETA optimizer outperforms our base optimizers and shows advantages over recently proposed competitive optimizers. We also provide theoretical proof of these two algorithms, which verifies the convergence of our proposed ADMETA.

## 1 INTRODUCTION

The field of training neural network is dominated by gradient decent optimizers for a long time, which use first order method. Typical ones include SGD (Robbins & Monro, 1951) and SGD with momentum (SGDM) (Sutskever et al., 2013), which are simple yet efficient algorithms and enjoy even better resulting convergence than many recently proposed optimizers. However, it suffers the disadvantage of low speed in initial stage and poor performance in sparse training datasets. This shortcoming can not be ignored since with the development of deep learning, the amount of data becomes much larger, and the model becomes much more complex. Time to train a network is also considered an important metric when evaluating an optimizer. To address this issue, optimizers with adaptive learning rate have been proposed which use nonuniform stepsizes to scale the gradient while training, and the usual implementation is scaling the gradient by square roots of some kind of combination of the squared values of historical gradients. By far the most used are Adam (Kingma & Ba, 2014) and AdamW (Loshchilov & Hutter, 2017) due to their simplicity and high training speed in early stage. Despite their popularity, Adam and many variants like of it (such as RAdam (Liu et al., 2019)) is likely to achieve worse generalization ability than non-adaptive optimizers, observing that their performance quickly plateaus on validation sets.

To achieve a better tradeoff, researchers have made many improvements based on SGD and Adam family optimizers. One attempt is switching from adaptive learning rate methods to SGD, based on the idea of complementing each other's advantages. However, a sudden change from one optimizer to another in a set epoch or step is not applicable because different algorithms make characteristic choices at saddle points and tend to converge to final points whose loss functions nearby have different geometry (Im et al., 2016). Therefore, many optimizers based on this idea seek for a smooth switch. The representative ones are AdaBound (Luo et al., 2019) and SWATS (Keskar & Socher, 2017). The second attempt is proposing new method to further accelerate SGDM, including introducing power

exponent (pbSGD (Zhou et al., 2020)), aggregated momentum (AggMo (Lucas et al., 2018)) and warm restarts (SGDR (Loshchilov & Hutter, 2016)). The third attempt is modifying the process of optimizers with adaptive learning rate to achieve better local optimum, which is the most popular field in recent researches (Zhuang et al., 2020; Li et al., 2020). Due to space constraints, please see more related work in Appendix A.

We focus in this paper on the use of historical and future information about the optimization process of the model, both of which we argue are important for models to reach their optimal points. To this end, we introduce a bidirectional view, backward-looking and forward-looking. In the backward-looking view, EMA is an exponentially decreasing weighted moving average, which is used as a trend-type indicator in terms of the optimization process. And since the training uses a mini-batch strategy, each batch is likely to have deviations from the whole, so it may mislead the model to the local optimal point. Inspired by stock market indicators, DEMA (Mulloy, 1994) is an exponential average calculated on the traditional EMA and current input, which can effectively maintain the trend while reducing the impact caused by short-term bias. We thus replace the traditional exponential moving average (EMA) with double exponential moving average (DEMA). It is worth noting that our usage is not equivalent to the original DEMA, but rather a variant of it. In the forward-looking part, since we observe that a constant weight adopted by the original Lookahead optimizer (Zhang et al., 2019) to control the scale of fast weights and slow weights in each synchronization period makes the early stage training slow and lossy, we propose a new dynamic strategy which adopts an asymptotic weight for improvement. By applying these two ideas, we propose ADMETA optimizer with ADMETAR and ADMETAS implementations based on RAdam and SGDM respectively.

Extensive experiments have been conducted on computer vision (CV), natural language processing (NLP) and audio processing tasks, which demonstrate that our method achieves better convergence results compared to other recently proposed optimizers. Further analysis show that ADMETAS achieves higher generalization ability than SGDM and ADMETAR achieves better convergence results and maintain high speed in initial stage compared to other adaptive learning rate methods. We further find that DEMA and dynamic looking strategy can improve performance compared to EMA and constant strategy, respectively. In addition, we provide convergence proof of our proposed ADMETA in convex and non-convex optimizations.

## 2 ADMETA

### 2.1 BACKGROUND

The role of the optimizer in model training is to minimize the loss on the training set and thus drive the learning of model parameters. Formally, consider a loss function $f : \mathbb{R}^d \to \mathbb{R}$ that is bounded below greater than zero, where $\mathbb{R}$ represents the field of real numbers, $d$ denotes the dimension of the parameter and thus $\mathbb{R}^d$ denotes d-dimensional Euclidean space. The optimization problem can be formulated as: $\min_{\theta \in \mathcal{F}^d} f(\theta)$, where $\theta$ indicates a parameter whose domain is $\mathcal{F}$ and $\mathcal{F} \subset \mathbb{R}^d$. If we define the optimum parameter of the above loss function as $\theta^*$, then the optimization objective can be written as:

$$\theta^* = \arg\min_{\theta \in \mathcal{F}^d} f(\theta). \tag{1}$$

Optimizers iteratively update parameters to make them close to the optimum as training step $t$ increases, that is to make: $\lim_{t \to \infty} \|\theta_t - \theta^*\| = 0$.

Stochastic gradient algorithm SGD (Robbins & Monro, 1951) optimizes $f$ by iteratively updating parameter $\theta_t$ at step $t$ in the opposite direction of the stochastic gradient $g(\theta_{t-1}; \xi_t)$ where $\xi_t$ is the input variables of the $t$-th mini-batch in training datasets. For the sake of clarity, we abbreviate $g(\theta_{t-1}; \xi_t)$ as $g_t$ for the rest of the paper unless specified. SGD optimization aims to calculate the updated model parameters based on the previous model parameters, the current gradient and the learning rate. Define learning rate as $\alpha_t$, the update process is summarized as follows:

$$\theta_t = \theta_{t-1} - \alpha_t g_t. \tag{2}$$

Original SGD tend to vibrate along the process due to the mini-batch strategy and not using of past gradients. What's more, this disadvantage also results in its long-time plateaus in valleys and saddle points, thus slowing the speed. To smooth the oscillation and speed up convergence rate, momentum, also known as Polyak's Heavy Ball (Polyak, 1964), is introduced to modify SGD. Momentum at step

$t$ is often denoted as $m_t$ and obtained by iterative calculation with a dampening coefficient $\beta$. Thus, the update process of SGD with momentum (SGDM) (Sutskever et al., 2013) becomes as follows:

$$m_t = \beta m_{t-1} + (1 - \beta)g_t, \tag{3}$$
$$\theta_t = \theta_{t-1} - \alpha_t m_t, \tag{4}$$

Although momentum works well, the uniform stepsize on every parameter is also another factor to limit the speed, especially in large datasets and sparse datasets. To further accelerate the update, adaptive learning rate optimizer is introduced which adopts an individual stepsize for each parameter based on their unique update process.

Since a smoothing mechanism is employed in the calculation of stepsize, two dampening coefficients, $\beta_1$ and $\beta_2$, are introduced for balancing the current and historical information. Adam (Kingma & Ba, 2014), a typical adaptive learning rate optimizer, is implemented as follows:

$$m_t = \beta_1 m_{t-1} + (1 - \beta_1)g_t, \tag{5}$$
$$v_t = \beta_2 v_{t-1} + (1 - \beta_2)g_t^2, \tag{6}$$
$$\theta_t = \theta_{t-1} - \alpha_t m_t / \sqrt{v_t}, \tag{7}$$

where $m_t$ indicates the first momentum, corresponding to the momentum in SGDM; $v_t$ indicates the second momentum.

To emphasize the functionality of $v_t$, we call it adaptive item for the rest of the paper. Adam may sometimes converge to bad local optimum, partly due to its large variance in the early stage. To fix this issue, RAdam (Liu et al., 2019) introduces a further rectified item $r_t$ and split the update process into two sub-processes sequentially connected:

$$\rho_\infty = 2/(1 - \beta_2) - 1, \tag{8}$$
$$\rho_t = \rho_\infty - 2t\beta_2^t/(1 - \beta_2^t), \tag{9}$$
$$r_t \leftarrow \sqrt{\frac{(\rho_t - 4)(\rho_t - 2)\rho_\infty}{(\rho_\infty - 4)(\rho_\infty - 2)\rho_t}}, \tag{10}$$
$$\theta_t = \begin{cases} \theta_{t-1} - \alpha_t m_t, & \rho_t \leq 4 \\ \theta_{t-1} - \alpha_t r_t m_t / \sqrt{v_t}, & \rho_t > 4 \end{cases}. \tag{11}$$

## 2.2 BACKWARD-LOOKING

In fact, the calculation of momentum $m_t$ in Eq. (3) and Eq. (5) is an exponential moving average (EMA) on gradient $g_t$. EMA, also known as exponential weighted moving average, can be used to estimate the local mean value of variables, so that the update of variables is related to historical values over a period of time. Formally, EMA is expressed as:

$$S_t = \beta S_{t-1} + (1 - \beta)p_t, \tag{12}$$

where the variable $S$ is denoted as $S_t$ at time $t$ and $p_t$ is the new assigned values. Particularly, $S_t = p_t$ without using EMA. In Eq. (3), SGDM employs EMA to take a moving average of the past gradients. While in Eq. (5), Adam and RAdam further apply EMA on the square of past gradients to construct the adaptive item. In the EMA, the moving average of the variable $S$ at time $t$ is roughly equal to the average of the values $p$ over the past $1/(1 - \beta)$ steps. This makes the moving average vary more at the beginning, so a bias correction is proposed and used in Adam (Eq. (7)) and in RAdam (Eq. (11)) when $\rho > 4$.

EMA can be regarded as obtaining the average values of the variables over time. Compared with the direct assignment of values to variables, the change curve of the values obtained by moving average is smoother and less jittery, and the moving average does not fluctuate greatly when inputting outliers, which is very important for the optimization using sampled mini-batch. Although efficient, EMA is not necessarily the best strategy for using historical information when it comes to the backward-looking part. Although it can effectively suppress the vibration caused by mini-batch training by performing the moving average on $g_t$, it also brings a lag time that affects the convergence speed and increases with the length of the moving average. What's more, it can result in overshoot problem (An et al., 2018), one possible reason is that EMA might make the wrong use of historical gradients in the final stage and thus have a "burden" to converge to optimum.

Double Exponential Moving Average (DEMA), first proposed by Mulloy (1994), is a faster moving average strategy and was invented to reduce the lag time of EMA. Thus, motivated by the advantage of DEMA, we developed a DEMA variant for the model optimization. It is worth noting DEMA is not simply taking a moving average of historical gradients twice, instead, it takes the moving average of the linear combination of the current gradient the moving average of past gradients. The form of

our DEMA variant can be written as:

$$\mathbf{DEMA} = \mathbf{EMA}^{out}(\mu\mathbf{EMA}^{in} + \kappa g_t), \tag{13}$$

where $\mu$ and $\kappa$ are coefficients that control the scale of current gradient and only depends on $\beta$.

From the formula $\mathbf{EMA} = \Sigma_{i=1}^{n}\beta^{n-i}g_i$, past gradients follow a fixed proportionality, that is, the ratio of gradient weight at one time to gradient weight at the previous time is $\beta$.

Due to the use of minibatch training strategy, the input is randomly sampled. The effect of each minibatch towards optimization is varied. Therefore, applying a fixed proportional to past gradients is not a reasonable approach since it does not take into account the changeable situation. The disadvantage of overshoot that EMA usually has may also be caused by the above reasons (An et al., 2018).

Thus, we deal with the relationship between the historical gradients and the current gradient more flexibly by further controlling the proportion of past gradients. Our design of coefficients in DEMA is also for this purpose. Based on Eq. (13), our actual implementation on algorithm is:

$$I_t = \lambda I_{t-1} + g_t, \tag{14}$$
$$h_t = \kappa g_t + \mu I_t + \nu, \tag{15}$$
$$m_t = \beta m_{t-1} + (1-\beta)h_t, \tag{16}$$

where $I_t$ is the output of $\mathbf{EMA}^{in}$ with a 0 initial value and $m_t$ is the output of $\mathbf{EMA}^{out}$ also initiated with 0. $\lambda$ and $\beta$ are dampening coefficients of inner EMA and outer EMA respectively, $\nu$ is a bias item, which is set to a small amount that decreases exponentially to 0 and chosen as $\lambda^t g_1$. The bias item does not affect the convergence proof, so for the sake of brevity, it is omitted for the rest of this paper and the details can be seen in the code. Please refer to Appendix B for more comparison and discussion between EMA and DEMA.

## 2.3 FORWARD-LOOKING

Focusing on gradient history, that is, backward-looking, optimizer is conducive to alleviating the vibration problem in the optimization process and preventing it from being misled by local noise information. However, since the optimization problem of the deep neural network is very complex, the optimizer can make the optimization process more robust by pre-exploration, so as to obtain better optimization results, which is called forward-looking.

Based on Reptile algorithm and advances in understanding the loss surface, Zhang et al. (2019) proposed Lookahead optimizer, which introduces two update processes and averages fast and slow weights periodically. The algorithm can be expressed as the cycle of the following process:

**Pre-exploration** : $\theta_t = \text{OPTIM}(\theta_{t-1})$
**Synchronization** : (*every k steps*)
$$\phi_t = \phi_{t-k} + \eta(\theta_{t-1} - \phi_{t-k})$$
$$\theta_t = \phi_t$$

where $\text{OPTIM}(\cdot)$ denotes a chosen optimizer, $k$ denotes the synchronization period, or in other words, the period of forward-looking, $\phi_t$ denotes the slow weights , $\theta_t$ denotes the fast weight updated with a chosen optimizer, and $\eta$ is a constant coefficient controlling the proportion of slow weights and fast weights in each synchronization. Generally, the chosen optimizer can be arbitrary.

We can get an intuitive explanation of Lookahead optimizer from the pseudo code above: Guided by fast weight $\theta_t$, the slow weight $\phi_t$ updates by taking linear interpolation between itself and the fast weight. Every time the fast weight updates $k$ steps, the slow weight updates 1 step. The update direction of slow weight can be regarded as $\theta_t - \phi_t$ from the equation. Therefore, $\eta$ can also be interpreted as the stepsize of slow weight in each synchronization. In order not to be confused with the stepsize of fast weight, we rename the stepsize of slow weight as $stepsize_s$. The recommended value of $\eta$ in (Zhang et al., 2019) is 0.5 and 0.8.

In the original Lookahead optimizer implementation, the fast and slow optimization processes were synchronized according to a given period, and parameters are fused at a fixed ratio during synchronization. However, optimization is a continuous process. In different optimization stages, fast optimization steps have different guiding effects on parameters. We argue that using fixed $stepsize_s$ in each synchronization is not an optimal strategy, and may even lead to negative effects. For this consideration, we turns the constant $\eta$ into a $\eta_t$ that changes over step monotonously and asymptotically. Generally, $\eta_t$ is a function that starts from 1 and converges to a set value and depends only on the step $t$. In this setting, the proportion of slow weights increases and this part gradually

turns into the original Lookahead method. In other words, the slow weights in our method adopts a faster $stepsize_s$ at the beginning, and it asymptotically slows down as processing. Specifically, we define two asymptotic functions for $\eta_t$:

$$\eta_t = 0.5 * \left(1 + \frac{1}{0.01\sqrt{t} + 1}\right), \quad \eta_t = 0.8 * \left(1 + \frac{1}{0.1\sqrt{t} + 3.8}\right), \tag{17}$$

thus we call this as dynamic asymptotic lookahead. The two functions are designed to turn $\eta_t$ from 1 to 0.5 and 0.8 respectively. Notably, these asymptotic functions may not be the best. We just find that it works well and maybe future work can done to investigate a more suitable one. For the sake of clarity, we will use the latter one in the rest of the paper and the results of experiments trained from scratch are based on this function unless specified.

To illustrate the advantages of our dynamic lookhead strategy over no lookhead and the original constant lookahead, we give an optimization example in Figure 1. In region ①, which is around early stage, the direction of the update is relatively stable and a large $stepsize_s$ is needed. $\theta_1 \to \theta_4$ denotes the update of fast weights. A constant lookahead method will slow the update process in each synchronization period, as can be seen in $\theta_1 \to \theta_2$. In our method, fast weights share more proportion in each synchronization period in early stage, thus updating more fast, as can be seen in $\theta_1 \to \theta_3$.

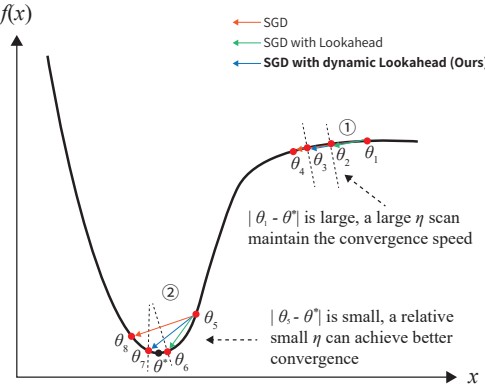

Figure 1: Comparison between no lookahead, constant lookahead and dynamic lookahead.

In region ②, which is around final stage, the direction of the update is relatively ocillated, and a small $stepsize_s$ is needed. fast weights tend to overshoot the optimum, as can be seen in $\theta_5 \to \theta_8$. Lookahead optimizer can achieve better convergence result than general algorithm as it averages the weights to make them more close to the optimum point, as can be seen in $\theta_5 \to \theta_6$. In our method, the proportion of fast weights have already reduced asymptotically to a set value, thus can achieve similar efficacy as Lookahead optimizer as can be seen in $\theta_5 \to \theta_7$.

From these analysis, we demonstrate that our dynamic lookhead strategy method improves the robustness of training.

### 2.4 IMPLEMENTATIONS OF ADMETAR AND ADMETAS

Since optimizers of the Adam family and SGD family have their own advantages and disadvantages, and the bidirectional looking optimizer framework and improvement we propose do not have too many restrictions on the basic optimizer, we have implemented improved versions ADMETAR and ADMETAS based on RAdam and SGDM optimizer. The final algorithm form is shown in Algorithm 1 and 2. Detailed proof of convergence and convergence rate for our ADMETAR and ADMETAS is putted in Appendix C and D.

## 3 EXPERIMENTS

In this section, we demonstrate the effectiveness of our optimizer by turning to an empirically exploration of different datasets and different models to compare some popular optimizers. Specifically, we conduct experiments on typical CV, NLP, and audio processing tasks. Influenced by the Transformer structure, models are becoming deeper and larger, and therefore training is becoming more difficult. The current paradigm of pre-training-fine-tuning is mainly used for large models. Therefore, we compare optimizers not only in the training-from-scratch setup, but also in the fine-tuning setup.

In this section, we compare our proposed optimizer with several typical optimizers, including classic SGD (Robbins & Monro, 1951) and Adam (Kingma & Ba, 2014), our base, SGDM (Sutskever et al., 2013)[1] and RAdam (Liu et al., 2019), the current state-of-the-art AdaBelief (Zhuang et al., 2020),

---

[1]Notably, we employed nesternov momentum (Nesterov, 1983) in the SGDM for a stronger comparison baseline

and the optimizer combined of many modules, Ranger (Wright, 2019). Please refer to Appendix E for more experimental details.

---

**Algorithm 1:** ADMETAR Optimizer. All operations are element-wise.

---

**Initialize** $\theta_1 \in \mathcal{F}$, $\phi_0 \leftarrow 0$, $m_0 \leftarrow 0$, $v_0 \leftarrow 0$, $I_0 \leftarrow 0$, $t \leftarrow 0$
**for** t=1,2,... **do**

    $t \leftarrow t + 1$
    $g_t \leftarrow \nabla_t f_t(\theta_t)$
    $I_t \leftarrow \lambda I_{t-1} + g_t$
    $h_t \leftarrow \kappa g_t + \mu I_t$
    $m_t \leftarrow \beta_1 m_{t-1} + (1 - \beta_1) h_t$
    $\rho_t \leftarrow \rho_\infty - 2t \frac{\beta_2^t}{1 - \beta_2^t}$

    **if the variance is tractable, i.e.,**
    $\rho_t > 4$**, then**

        $v_t \leftarrow \beta_2 v_{t-1} + (1 - \beta_2) h_t^2$
        $r_t \leftarrow \sqrt{\frac{(\rho_t - 4)(\rho_t - 2)\rho_\infty}{(\rho_\infty - 4)(\rho_\infty - 2)\rho_t}}$
        $\widehat{m_t} \leftarrow \frac{m_t}{1 - \beta_1^t}$, $\widehat{v_t} \leftarrow \frac{v_t}{1 - \beta_2^t}$
        $\theta_{t+1} \leftarrow \prod_{\mathcal{F}, \sqrt{\widehat{v_t}}}(\theta_t - \alpha_t \frac{r_t \widehat{m_t}}{\sqrt{\widehat{v_t}} + \epsilon})$

    **else**

        $\theta_{t+1} \leftarrow \prod_{\mathcal{F}, \sqrt{\widehat{v_t}}}(\theta_t - \alpha_t \widehat{m_t})$

    **if** t+1 % k == 0:

        $\phi_t \leftarrow \eta_t \theta_t + (1 - \eta_t)\phi_{t-k}$
        $\theta_t \leftarrow \phi_t$

**end for**
**return** $x$

---

**Notations:**

- $\alpha_t$: learning rate at step t
- $\lambda, \beta, \beta_1, \beta_2$: the momentum coefficients
- $\epsilon$: a small value used to avoid a zero denominator
- $k$: synchronization period
- $\prod_{\mathcal{F}, M}(y) = \text{argmin}_{x \in \mathcal{F}} ||M^{1/2}(x - y)||$
- $\mu = 25 - 10\left(\lambda + \frac{1}{\lambda}\right), \kappa = \frac{10}{\lambda} - 9$

---

**Algorithm 2:** ADMETAS Optimizer. All operations are element-wise.

---

**Initialize** $\theta_1 \in \mathcal{F}$, $\phi_0 \leftarrow 0$, $m_0 \leftarrow 0$, $I_0 \leftarrow 0$, $t \leftarrow 0$
**for** t=1,2,... **do**

    $t \leftarrow t + 1$
    $g_t \leftarrow \nabla f_t(\theta_t)$
    $I_t \leftarrow \lambda I_{t-1} + g_t$
    $h_t \leftarrow \kappa g_t + \mu I_t$
    $m_t \leftarrow \beta m_{t-1} + (1 - \beta) h_t$
    $\theta_{t+1} \leftarrow \theta_t - \alpha_t m_t$
    **if** t+1 % k == 0:

        $\phi_t \leftarrow \eta_t \theta_t + (1 - \eta_t)\phi_{t-k}$
        $\theta_t \leftarrow \phi_t$

**end for**
**return** $x$

---

### 3.1 IMAGE CLASSIFICATION

Consistent with general optimizer researches (Zhuang et al., 2020), we conduct experiments on two image classification tasks, CIFAR-10 and CIFAR-100 (Krizhevsky et al., 2009) in CV field, and the results are presented in Table 1. For model baselines, we choose the popular and leading performance ResNet-110 (He et al., 2016) and PyramidNet (Han et al., 2017), respectively. From the experimental results, whether in CIFAR-10 or CIFAR-100 dataset, and based on the ResNet-110 or PyramidNet model, SGDM achieves better results than SGD, indicating that backward-looking improves the optimization effect. EMA with rectified item in RAdam performs better than EMA in Adam, suggesting that a better backward-looking process can lead to performance gains. Comparing SGDM and RAdam, we find that SGDM has a performance advantage, showing that though Adam uses an adaptive learning rate to improve the speed of convergence, it is lossy for performance.

Among optimizers with adaptive learning rate, AdaBelief achieves better results than Adam and RAdam in CIFAR-10 with PyramidNet and CIFAR-100 with ResNet-110 and PyramidNet. Ranger, which combines forward and backward looking, achieves better performance than the backward-looking-only RAdam in CIFAR-10 and CIFAR-100 with PyramidNet. Our ADMETAR achieves consistent improvement over the optimizer baseline RAdam, which also confirms the gain of bidirectional looking for optimization. And ADMETAR has better results than Ranger, indicating that our bidirectional looking is better than Ranger's simple combination of multiple optimization features. Our ADMETAS also performs better than SGDM, further demonstrating the adaptability of our approach, which not only performs well in Adam family, but also works in SGD family.

Following the previous practice (Liu et al., 2019), we visualize the optimization process of the ResNet-110 model with Adam, RAdam, SGDM, and our ADMETAS, ADMETAR optimizers on the CIFAR-10 and CIFAR-100 datasets in Figure 2. As can be seen from the training loss figure, the above optimizers can successfully train the model to converge to a stable state, but ADMETAS obtains the lowest training loss on CIFAR-10, while AdaBelief obtains the training loss on CIFAR-100. In terms of performance on the test set, ADMETAS has obtained the best generalization ability,

Table 1: Results on CIFAR-10 and CIFAR-100 test sets.

| Model | CIFAR-10 | | CIFAR-100 | |
|---|---|---|---|---|
| | ResNet-110 | PyramidNet | ResNet-110 | PyramidNet |
| SGD | 90.27±0.15 | 91.52±0.03 | 65.70±0.25 | 76.51±0.06 |
| SGDM | 93.68±0.20 | 95.08±0.13 | 72.07±0.28 | 79.49±0.11 |
| Adam | 91.89±0.23 | 94.55±0.24 | 68.45±0.43 | 76.72±0.32 |
| RAdam | 93.09±0.05 | 94.58±0.14 | 70.39±0.08 | 76.02±0.53 |
| Ranger | 92.85±0.34 | 94.76±0.03 | 68.96±0.68 | 76.35±0.08 |
| AdaBelief | 92.81±0.26 | 94.70±0.03 | 70.88±0.07 | 76.57±0.04 |
| ADMETAR | **93.63±0.22** | **94.81±0.19** | **71.00±0.05** | **76.82±0.07** |
| ADMETAS | **94.12±0.17** | **95.30±0.08** | **73.74±0.26** | **79.61±0.34** |

Figure 2: Training loss and test accuracy comparison on CIFAR-10 and CIFAR-100 datasets.

which shows that the lower the loss of the training set may not necessarily lead to the better the generalization ability of the model. In addition, from the accuracy of the test set, the convergence speed of the SGD family including SGDM and ADMETAS is generally slower than that of the Adam family (Adam, RAdam, Ranger, AdaBelief and ADMETAR), but the final convergence result of the SGD family is better than the Adam family. However, our ADMETAR achieve more comparable performance to the SGD family, while maintaining the advantage of the fast convergence of the Adam family. ADMETAR has the highest results on the test set in the early stage of optimization ($< 80$ epoch), which demonstrates that bidirectional looking improves both accuracy and speed, making ADMETAR a efficient and effective optimizer implementation.

Compared to ResNet-110, PyramidNet has a more complicated structure and can achieve better results in these tasks. In cases where the model is strong enough, the selection of optimizer will not be the main factor for the final performance. As shown in Table 1, compared to Adam, RAdam and AdaBelief achieve just a bit of improvement on CIFAR-10 task and even achieve worse results on CIFAR-100 task, which also verifies our above claims.

## 3.2 NATURAL LANGUAGE UNDERSTANDING

As a general AI component, the general capability requirement for various tasks and various models is a basic requirements for optimizers. We evaluate the adaptability of our ADMETA optimizer on the finetune training scenario with current popular pre-trained language models. Since the SGD family converges slowly on the finetune stage of the Transformer architecture, we only compare the various optimizers of the Adam family here. Specifically, we conduct experiments based on the pre-trained language model BERT (Devlin et al., 2018) on three natural language understanding tasks, GLUE benchmark (Wang et al., 2018), machine reading comprehension (SQuAD v1.1 and v2.0 (Rajpurkar et al., 2016)) and named entity recognition (NER-CoNLL03 (Sang & De Meulder, 2003)). We report results for two model sizes, $BERT_{base}$ and $BERT_{large}$ to explore whether model size has an effect on the optimizer.

In Table 2, we report the results of on the development set of 8 datasets of the GLUE benchmark, where *Acc*, MCC, SCC are abbreviations of accuracy, Matthews Correlation and Spearman Correlation Coefficient, respectively. First, under the BERT-base model, compared with the basic optimizer RAdam, ADMETAR achieves consistent improvement. The most significant improvement is obtained on RTE and CoLA, which indicates that our ADMETA optimizer exhibits greater stability for low-resource optimization. On the other seven datasets, some of them are slightly improved. This is

Table 2: Development results on GLUE benchmark.

| Model | Optim | MNLI m/mm (*Acc*) | QQP (*F*$_1$) | QNLI (*Acc*) | SST-2 (*Acc*) | CoLA (*MCC*) | STS-B (*SCC*) | MRPC (*F*$_1$) | RTE (*Acc*) | Average |
|---|---|---|---|---|---|---|---|---|---|---|
| BERT$_{base}$ | AdamW | 83.85/84.08 | 87.72 | 90.74 | 93.23 | 60.32 | 89.11 | 90.85 | 67.51 | 82.92 |
| | Ranger | 83.80/84.24 | 87.83 | 90.76 | 92.32 | 58.87 | 89.19 | 90.05 | 68.59 | 82.68 |
| | AdaBelief | **83.91**/84.42 | 86.76 | 90.92 | 92.55 | 58.05 | 88.94 | 90.38 | 67.87 | 82.42 |
| | RAdam | **83.91**/84.24 | 87.66 | 90.88 | 92.20 | 59.31 | 89.07 | 90.91 | 70.04 | 83.00 |
| | ADMETAR | 83.90/**84.53** | **87.91** | **91.14** | **93.35** | **62.07** | **89.62** | **91.47** | **71.48** | **83.87** |
| BERT$_{large}$ | AdamW | 86.05/86.58 | **88.58** | 92.40 | 93.00 | 59.58 | 89.21 | 91.67 | 71.12 | 83.95 |
| | Ranger | **86.53**/86.58 | **88.58** | 92.39 | 93.46 | 63.81 | 89.73 | 92.04 | 72.56 | 84.89 |
| | AdaBelief | 85.59/86.25 | 86.99 | 92.42 | 93.00 | 61.11 | **90.17** | 91.28 | 72.92 | 84.19 |
| | RAdam | 86.40/**86.72** | 88.36 | 92.35 | **93.69** | 62.61 | 89.64 | 91.29 | 71.48 | 84.48 |
| | ADMETAR | 86.21/86.54 | 88.54 | **92.63** | **93.69** | **64.12** | 89.92 | **92.10** | **73.65** | **85.11** |

Table 3: Results on SQuAD v1.1 and v2.0 development sets and NER-CoNLL03 test sets.

| Model | Optim | SQuAD v1.1 | | SQuAD v2.0 | | NER-CoNLL03 | | |
|---|---|---|---|---|---|---|---|---|
| | | *EM* | *F*$_1$ | *EM* | *F*$_1$ | *P* | *R* | *F*$_1$ |
| BERT$_{base}$ | AdamW | 80.87 | 88.39 | 72.63 | 75.99 | 94.65 | 95.24 | 94.94 |
| | Ranger | 81.30 | 88.58 | 73.32 | 76.73 | 94.47 | 95.17 | 94.82 |
| | AdaBelief | 80.63 | 88.10 | 72.97 | 76.25 | 93.79 | 94.60 | 94.19 |
| | RAdam | 80.68 | 88.19 | 73.21 | 76.49 | 94.61 | 95.42 | 95.01 |
| | ADMETAR | **81.55** | **88.69** | **73.81** | **77.19** | **94.96** | 95.41 | **95.13** |
| BERT$_{large}$ | AdamW | 83.31 | 90.39 | 76.67 | 80.02 | 94.77 | 95.73 | 95.24 |
| | Ranger | 84.21 | **90.97** | 77.22 | 80.35 | 95.24 | 95.89 | 95.56 |
| | AdaBelief | 83.53 | 90.42 | **77.48** | 80.57 | 94.28 | 95.17 | 94.72 |
| | RAdam | 84.17 | 90.90 | 77.39 | **80.72** | 94.80 | 95.64 | 95.22 |
| | ADMETAR | **84.25** | 90.92 | 77.08 | 80.36 | 95.38 | 95.93 | **95.65** |

because most of the parameters of the model in the pre-training-fine-tuning paradigm have converged to a certain extent in the pre-training stage, so the further advantage of the optimizer in finetune is not apparent. And when the model is switched to a larger BERT-large, most tasks receive performance gains, except for CoLA and RTE using AdamW optimizer. Due to the further increase in model parameters, the low-resource dataset is not enough to fine-tune the large model, it will even reduce the model performance. But RAdam with rectified item, Ranger with bidirectional looking, and our ADMETAR handle the low-resource challenge well, continue to improve performance, and take advantage of large models. Our ADMETAR achieves the best results on these two low-resource datasets, demonstrating the effectiveness of our bidirectional looking approach.

In Table 3, we further report the results of machine reading comprehension and named entity recognition. ADMETAR achieved improvements at both model sizes in SQuAD v1.1 dataset, while similar improvements were achieved in SQuAD v2.0 with more complex models, illustrating that our optimizer is model-independent. Named entity recognition has reached a very accurate level with the help of pre-trained language models, and our ADMETAR optimizer also brings performance improvements over such a strong baseline, showing that optimization is also a bottleneck that restricts further performance improvement in addition to model structure and data.

## 3.3 AUDIO CLASSIFICATION

Like images and natural language, speech is one of the mainstream fields of deep learning research. In speech processing, there are also a large number of pre-trained large models, such as Wav2vec (Schneider et al., 2019). To highlight the input-independent nature of the optimizer, we also conduct experiments on two typical tasks of audio classification, keyword spotting (SUPERB) (Yang et al., 2021) and language identification (Common Language) (Sinisetty et al., 2021). We employ Wav2vec 2.0$_{base}$ as the baseline model and report the results of each optimizer in Table 4. In addition, we also list the training time of each optimizer to evaluate the impact of the bidirectional looking mechanism on the optimizer time overhead[2].

ADMETAR shows better classification accuracy than AdamW, RAdam, Ranger and AdaBelief, which is consistent with the experimental conclusions in image and natural language tasks. Consistent

---

[2]Notably, the reported training time is only for rough comparison due to the influence of environments.

Table 4: Results on speech keyword spotting and language identification tasks.

| Optim | SUPERB | | Common Language | |
|---|---|---|---|---|
| | *Acc* | Training | *Acc* | Training |
| AdamW | 98.26 | 10m44s | 79.45 | 8h27m33s |
| AdaBelief | 98.41 | 11m20s | 80.29 | 8h28m25s |
| Ranger | 98.35 | 11m50s | 81.18 | 8h29m55s |
| RAdam | 98.37 | 11m30s | 80.35 | 8h28m38s |
| ADMETAR | **98.50** | 11m54s | **81.57** | 8h30m15s |

Table 5: Ablation study on ADMETA optimizer.

| Optim | CIFAR-10 | Δ | Optim | CIFAR-10 | Δ |
|---|---|---|---|---|---|
| Adam | 91.89 | \ | SGD | 90.22 | \ |
| RAdam | 93.09 | \ | SGDM | 93.68 | \ |
| ADMETAR | **93.63** | | ADMETAS | **94.12** | |
| -DEMA | 93.24 | -0.39 | -DEMA | 89.13 | -4.99 |
| -LB | 92.29 | -0.95 | -LB | 89.88 | -4.24 |
| -LF | 93.14 | -0.10 | -LF | 93.51 | -0.61 |
| -LB-LF | 92.36 | -0.88 | -LB-LF | 89.80 | -4.32 |
| ADMETAR w/ constant LF | 93.03 | -0.60 | ADMETAS w/ constant LF | 93.75 | -0.37 |

results across image, natural language, and speech modalities verify the task-independence of our optimizer. Comparing the training time of ADMETAR with AdamW, RAdam, Ranger, and AdaBelief, our ADMETAR have different degrees of increase due to the additional computation and storage in the optimization process. Ranger and our ADMETAR increased the time most, but it can still be regarded as slight compared to the overall training time. Therefore, it can be concluded that the bidirectional looking mechanism adopted by ADMETA optimizer will bring additional computational overhead and increase the training time, but compared with the overall training cost, it is very small. ADMETA achieves better performance without increasing model parameters and training data, and does not have any impact on the inference time of the model, which achieves a better tradeoff.

## 4 ABLATION STUDY

We perform ablation study on various designs of ADMETA in bidirectional looking in this section. -DEMA means removing the DEMA mechanism in backward-looking and using the original EMA. -LB means complete removal of backward-looking, -LF means complete removal of forward-looking. -LB-LF means to remove bidirectional looking at the same time. w/ constant LF means use the original Lookahead mechanism in the forward-looking. The results are evaluated using the ResNet-110 model on the test set of CIFAR-10. According to the results shown in Table 5, it can be found that the improvement of SGDM compared with SGD initially shows the advantage of backward-looking. And compared with Adam, RAdam reveals that the EMA with the rectified item in backward-looking is more suitable for the training of the model than the original EMA. Our ADMETA (including ADMETAR and ADMETAS) achieved the best results. After removing DEMA and replacing dynamic lookahead with constant lookahead, respectively, the performance drops, indicating that both DEMA and dynamic asymptotic lookahead play an important role in stable optimization. After further removing the backward-looking, the forward-looking, and the bidirectional looking, the results drop further, validating our argument that bidirectional looking is beneficial for optimization.

## 5 CONCLUSION

In this paper, we introduce a bidirectional looking optimizer framework, exploring the use of historical and future information for optimization. For backward-looking, we introduce a DEMA scheme to replace the traditional EMA strategy, while for forward-looking, we propose a dynamic asymptotic lookahead strategy to replace the constant lookahead scheme. In this way, we propose ADMETA optimizer, and provide two implement versions, ADMETAR and ADMETAS, which are based on adaptive and non-adaptive momentum optimizers, RAdam and SGDM respectively. We verify the benefits of ADMETA with intuitive examinations and various experiments, showing the effectiveness of our proposed optimizer. Please refer to Appendix F for future work discussion.

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

APPENDIX

## A   RELATED WORK

As an important part of machine learning and deep learning, optimizer has received much attention in recent years. The optimizer plays a prominent role in the convergence speed and the convergence effect of the model. To seek good properties like fast convergence, good generalization and robustness, many algorithms have been put forward recently, and they can be divided into four families according to their characteristics and motivation.

**SGD Family**   In this family, the optimizers adopt the method of update like

$$\theta_t = \theta_{t-1} - \alpha_t m_t,$$

where $\theta_t$ denotes the parameter to be optimized at iteration step t and $m_t$ refers to some combination of past gradients (such as EMA), which can be represented as $f_1(g_1, g_2, ..., g_t)$. Original SGD (Robbins & Monro, 1951) directly minus the product of global learning rate and the gradient at each step. Despite of its simplicity, it is still widely used in many datasets. However, SGD is blamed for its low convergence rate and high fluctuation, thus many methods have been proposed to accelerate the speed and smooth the update process. One efficient optimizer to tackle this issue is SGDM (Sutskever et al., 2013), which uses a exponential moving average (EMA, also known as momentum) to replace the gradient with an exponential weight decay of past gradients. SGDM-Nesterov (Nesterov, 1983) is a variant of SGDM which modifies the momentum by computing gradient based on the approximation of the next position and thus changing the descent direction. Experiments have shown that Nesterov momentum tends to achieve a higher speed and performance.

**Adam Family**   The Adam family optimizers usually update parameters by

$$\theta_t = \theta_{t-1} - \alpha_t m_t / \sqrt{v_t},$$

where $v_t$ is the adaptive item and can be represented as $f_2(g_1^2, g_2^2, ..., g_t^2)$. Compared to SGD family, instead of using a uniform learning rate, this kind of optimizer computes an individual learning rate for each parameter due to the effect of the denominator $\sqrt{v_t}$ in the equation. $v_t$ is usually an dimension-reduction approximation to the matrix which contains the information of second order curvature, such as Fisher matrix (Pascanu & Bengio, 2013).

Adadelta Zeiler (2012), Adagrad (Duchi et al., 2011) and RMSprop (Tieleman & Hinton, 2012) are early optimizers in this family. A stand out generation is Adam (Kingma & Ba, 2014) which combines the RMSprop with Adagrad. It has been widely used in a wide range of datasets and works well even with sparse gradients. However, there are problems with Adam with respect to convergence and generalization, thus many methods have been proposed to make improvements

Based on the large variance in the early stage that may leads to a bad optimum, heuristic warmup (Vaswani et al., 2017; Popel & Bojar, 2018) and RAdam (Liu et al., 2019) are proposed, of which the former starts with a small initial learning rate and the latter introduces a rectified item. To fix the convergence error, Reddi et al. (2019) proposed AMSGrad which requires the non-decreasing property of the second momentum. In fact, this method can be interpolated into other Adam family algorithms to guarantee the convergence in convex situations. Considering curvature of the loss function, AdaBelief (Zhuang et al., 2020) and AdaMomentum (Wang et al., 2021) are proposed. More recently, there are still numerous studies devoted to improving Adam, such as AdaX (Li et al., 2020) and AdaFamily (Fassold, 2022). However, we notice that most researches put a solid emphasis on modifying the second momentum term, i.e., the adaptive item and ignores the possibility to make a relative overall change to the algorithms.

**Stochastic Second-Order Family**   In the stochastic second-order optimizers, parameters are updated using second-order information related to Hessian matrix. The update process is typically written as

$$\theta_t = \theta_{t-1} - \alpha_t H^{-1} m_t,$$

where $H$ is the Hessian matrix or approximation matrix to it. Ideally, they can achieve better results than the first order optimizers (like Adam family and SGD family), but their practicality is limited

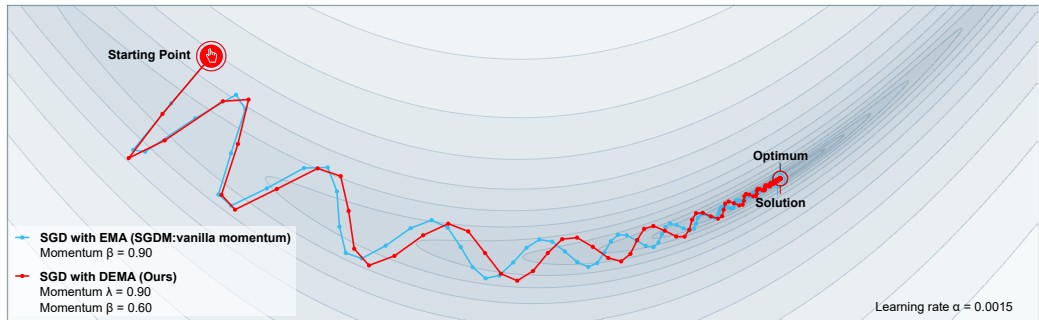

Figure 3: EMA vs. DEMA in SGD optimizer. Please refer to our online demo `https://sites.google.com/view/optimizer-admeta` for more comparison.

due to the large computational cost of the second order information, like the fisher / hessian matrix. Some methods have been proposed using low-rank decomposition and approximating to hessian diagonal to reduce the cost, like Apollo (Ma, 2020), AdHessian (Yao et al., 2021) and Shampoo (Anil et al., 2020).

**Other Optimizers**    There are some algorithms that are not convenient to be categorized into the above families and we list some examples here. Motivated by PID controller, SGD-PID (An et al., 2018) takes an analogy between gradient and the input error in a automatic control system. Analysis show that it can reduce the overshoot problem in SGD and SGD variants. Furthermore, Weng et al. (2022) applied PID into Adam and proposed AdaPID optimizer.

Lookahead (Zhang et al., 2019) optimizer updates two sets of weight wherein "fast weights" function as a guider to search for the direction and "slow weights" follows the guide to achieve better optimization. Ranger (Wright, 2019) optimizer further combines RAdam and Lookahead to get a compound algorithm and shows a better convergence performance.

**Discussion**    To show the advantage of bidirectional looking, we propose ADMETA optimizer. Specifically, it is based on the idea of considering backward-looking and forward-looking, wherein DEMA plays a important role in the former aspect and dynamic asymptotic forward-looking strategy serves for the latter aspect.

In practical use, we provide two versions, ADMETAS and ADMETAR, using the framework of ADMETA and based on SGDM and RAdam respectively. Specifically, ADMETAS replace the traditionally used EMA in backward-looking part of SGDM with DEMA and add the forward-looking part which is derived from Lookahead optimizer. ADMETAR is based on RAdam in the same way. The second order family is also introduced above because the framework of ADMETA can also be applied in this family, and it is remained as the future work.

## B   EMA VS. DEMA

To corroborate our analysis of EMA and DEMA, we compared the optimization process of EMA and DEMA on the SGD optimizer according to the practice of (Goh, 2017). Using the same learning rate $\alpha$ and starting from the same starting point, the convergence process is shown in Figure 3. The decent surface in the figure is the convex quadratic, which is a useful model despite its simplicity, for it comprises an important structure, the "valleys", which is often studied as an example in momentum-based optimizers. As demonstrated in Figure 3, on the one hand, DEMA achieves faster speed than EMA, which can be easily seen by comparing the distance to the optimal point at the same time; on the other hand, DEMA achieves better convergence results than EMA as can be seen in the distance between the point of convergence and optimum.

## C PROOF OF CONVERGENCE

In this section, following (Chen et al., 2018), (Alacaoglu et al., 2020) and (Reddi et al., 2019), we provide detailed proofs of convergence for ADMETAR and ADMETAS optimizers in convex and non-convex situations.

### C.1 CONVERGENCE ANALYSIS IN CONVEX AND NON-CONVEX OPTIMIZATION

**Optimization problem** For deterministic problems, the problem to be optimized is $\min_{\theta \in \mathcal{F}} f(\theta)$, where $f$ denotes the loss function. For online optimization, the problem is $\min_{\theta \in \mathcal{F}} \sum_{t=1}^{T} f_t(\theta)$, where $f_t$ is the loss function of the model with the given parameters at the $t$-th step.

The criteria for judging convergence in convex and non-convex cases are different. For convex optimization, following (Reddi et al., 2019), the goal is to ensure $R(T) = o(T)$, i.e., $\lim_{T \to \infty} R(T)/T = 0$. For non-convex optimization, following (Chen et al., 2018), the goal is to ensure $\min_{t \in [T]} \mathbb{E} \left|\left| \nabla f(\theta_t) \right|\right|^2 = o(T)$.

**Theorem C.1.** *(Convergence of* ADMETAR *for convex optimization)*
*Let $\{\theta_t\}$ be the sequence obtained from* ADMETAR, $0 \leq \lambda, \beta_1, \beta_2 < 1$, $\gamma = \frac{\beta_1^2}{\beta_2} < 1, \alpha_t = \frac{\alpha}{\sqrt{t}}$ *and $v_t \leq v_{t+1}, \forall t \in [T]$. Suppose $x \in \mathcal{F}$, where $\mathcal{F} \subset \mathbb{R}^d$ and has bounded diameter $D_\infty$, i.e. $||\theta_t - \theta||_\infty \leq D_\infty, \forall t \in [T]$. Assume $f(\theta)$ is a convex function and $||g_t||_\infty$ is bounded. Denote the optimal point as $\theta$. For $\theta_t$ generated,* ADMETAR *achieves the regret:*

$$R(T) = \sum_{t=1}^{T} [f_t(\theta_t) - f_t(\theta)] = \mathcal{O}(\sqrt{T})$$

**Theorem C.2.** *(Convergence of* ADMETAR *for non-convex optimization)*
*Under the assumptions:*

- *$\nabla f$ exits and is Lipschitz-continuous,i.e, $||\nabla f(x) - \nabla f(y)|| \leq L||x - y||$, $\forall x, y$; $f$ is also lower bounded.*

- *At step $t$, the algorithm can access a bounded noisy gradient $g_t$, and the true gradient $\nabla f$ is also bounded.*

- *The noisy gradient is unbiased, and has independent noise, i.e. $g_t = \nabla f(\theta_t) + \delta_t, \mathbb{E}[\delta_t] = 0$ and $\delta_i \perp \delta_j, \forall i \neq j$.*

*Assume $\min_{j \in [d]} (v_1)_j \geq c > 0$ and $\alpha_t = \alpha/\sqrt{t}$, then for any T we have:*

$$\min_{t \in [T]} \mathbb{E} \left|\left| \nabla f(\theta_t) \right|\right|^2 \leq \frac{1}{\sqrt{T}} (Q_1 + Q_2 log T)$$

*where $Q_1$ and $Q_2$ are constants independent of T.*

**Theorem C.3.** *(Convergence of* ADMETAS *for convex optimization)*
*Let $\{\theta_t\}$ be the sequence obtained by* ADMETAS, $0 \leq \lambda, \beta < 1$, $\alpha_t = \frac{\alpha}{\sqrt{t}}$, $\forall t \in [T]$. Suppose $x \in \mathcal{F}$, where $\mathcal{F} \subset \mathbb{R}^d$ and has bounded diameter $D_\infty$, i.e. $||\theta_t - \theta||_\infty \leq D_\infty, \forall t \in [T]$. Assume $f(\theta)$ is a convex function and $||g_t||_\infty$ is bounded. Denote the optimal point as $\theta$. For $\theta_t$ generated,* ADMETAS *achieves the regret:*

$$R(T) = \sum_{t=1}^{T} [f_t(\theta_t) - f_t(\theta)] = \mathcal{O}(\sqrt{T})$$

**Theorem C.4.** *(Convergence of* ADMETAS *for non-convex optimization)*
*Under the assumptions:*

- *$\nabla f$ exits and is Lipschitz-continuous,i.e, $||\nabla f(x) - \nabla f(y)|| \leq L||x - y||$, $\forall x, y$; $f$ is also lower bounded.*

- *At step $t$, the algorithm can access a bounded noisy gradient $g_t$, and the true gradient $\nabla f$ is also bounded.*

- *The noisy gradient is unbiased, and has independent noise, i.e.* $g_t = \nabla f(\theta_t) + \delta_t, \mathbb{E}[\delta_t] = 0$ *and* $\delta_i \perp \delta_j, \forall i \neq j$.

*Assume* $\alpha_t = \alpha/\sqrt{t}$, *then for any T we have:*

$$\min_{t \in [T]} \mathbb{E}\left\|\nabla f(\theta_t)\right\|^2 \leq \frac{1}{\sqrt{T}}(Q_1' + Q_2' log T)$$

*where* $Q_1'$ *and* $Q_2'$ *are constants independent of T.*

Before formally proving the theorems, here list some remarks and preparations.

**Remark 1.** For brevity, we omit the rectified item of ADMETAR in the proof. However, it does not influence the proof since it can be integrated into the learning rate.

**Remark 2.** Following (Luo et al., 2019), the bias correction $1/1 - \beta_1^t$ of the first momentum $m_t$ is omitted in the convergence of ADMETAR. Since $1/1 - \beta_1^t$ is bounded above 1 and below 10, the order of the terms used is not affected, thus hardly affecting the proof.

**Remark 3.** The forward-looking part is not considered in the proof. On the one hand, explanations and proofs of constant Lookahead have been given in (Zhang et al., 2019) and (Wang et al., 2020), which can be imitated by our dynamic method. On the other hand, forward-looking part is exactly the interpolation of fast weights and slow weights at each synchronization period, and the fast weights are updated by the given optimizer. Therefore, the convergence proof is equivalent to only proving convergence of fast weights.

**Lemma C.5.** *if* $\|g_t\|_\infty$ *is bounded,i.e.* $\|g_t\|_\infty \leq G_\infty, \forall t \in [T]$, *where* $G_\infty$ *is a constant independent of T, then* $I_t, h_t$ *and* $m_t$ *are also bounded.*

*Proof.* First of all, we prove $\|I_t\|_\infty \leq (1 + \lambda)G_\infty$ by induction:
when $t = 1$

$$\|I_1\|_\infty = \|g_1\|_\infty \leq G_\infty$$

Suppose $t = k$ satisfies, then for $t = k + 1$

$$\|I_{k+1}\|_\infty = \|\lambda I_k + g_{k+1}\|_\infty \leq \lambda\|I_k\|_\infty + \|g_{k+1}\|_\infty$$
$$\leq (\lambda + 1)max\{\|I_k\|_\infty, \|g_{k+1}\|_\infty\} \leq (1 + \lambda)G_\infty$$

Next, for $\|h_k\|_\infty$

$$\|h_t\|_\infty = \|\kappa g_t + \mu I_t\|_\infty \leq \kappa\|g_t\|_\infty + \mu\|I_t\|_\infty \leq [\kappa + (1 - \lambda)\mu)]G_\infty$$

Since $m_t$ is the moving average of $h_i$ where i=1,...,t, we can get that it is also bounded following the proof of $I_t$. $\qquad\square$

In this way, we can redefine $G_\infty$ by enlarging it and the bounded stochastic gradient assumption in the theorem is equivalent to assuming $\|g_t\|_\infty, \|I_t\|_\infty, \|h_t\|_\infty, \|m_t\|_\infty \leq G_\infty$.

**Remark 4.** As for non-convex optimization, in the same way, the bounded noisy gradient assumption is equivalent to $\|g_t\|, \|I_t\|, \|h_t\|, \|m_t\| \leq H$ where H is a constant independent of T. This remark will be used in several places in the following proof.

**Lemma C.6** (Generalized Hölder inequality, (Beckenbach & Bellman, 2012)). *For* $x, y, z \in \mathbb{R}_+^n$ *and positive* $p, q, r$ *such that* $\frac{1}{p} + \frac{1}{q} + \frac{1}{r} = 1$, *we have*

$$\sum_{j=1}^n \theta_j y_j z_j \leq \|x\|_p \|y\|_q \|z\|_r.$$

This is a common mathematical inequality, so the proof is omitted here.

**Lemma C.7** (nonexpansiveness property of $\arg\min_{x \in \mathcal{F}} \|.\|$, (McMahan & Streeter, 2010)). *For any* $Q \in \mathcal{S}_+^d$,*i.e.* $Q$ *is a Positive definite matrice and convex feasible set* $\mathcal{F} \subset \mathbb{R}^d$, *suppose* $u_1 = \arg\min_{x \in \mathcal{F}} \|Q^{1/2}(x - z_1)\|$ *and* $u_2 = \arg\min_{x \in \mathcal{F}} \|Q^{1/2}(x - z_2)\|$ *then we have* $\|Q^{1/2}(u_1 - u_2)\| \leq \|Q^{1/2}(z_1 - z_2)\|$.

*Proof.* First, we claim that $\langle u_1 - z_1, Q(u_2 - u_1)\rangle \geq 0$ and $\langle u_2 - z_2, Q(u_1 - u_2)\rangle \geq 0$ (We only prove the former as the proofs are exactly the same). Otherwise, consider a small $\delta$, we have $u_1 + \delta(u_2 - u_1) \in \mathcal{F}$

$$\frac{1}{2}\langle u_1 + \delta(u_2 - u_1) - z_1, Q(u_1 + \delta(u_2 - u_1) - z_1)\rangle$$

$$= \frac{1}{2}\langle u_1 - z_1, Q(u_1 - z_1)\rangle + \frac{1}{2}\delta^2\langle u_2 - u_1, Q(u_2 - u_1)\rangle + \delta\langle u_1 - z_1, Q(u_2 - u_1)\rangle$$

If there exists $\langle u_1 - z_1, Q(u_2 - u_1)\rangle < 0$, $\delta$ can be chosen so small that it satisfies $\frac{1}{2}\delta^2\langle u_2 - u_1, Q(u_2 - u_1)\rangle + \delta\langle u_1 - z_1, Q(u_2 - u_1)\rangle < 0$, which contradicts the definition of $u_1$.

Using the above claim, we further have

$$\langle u_1 - z_1, Q(u_2 - u_1)\rangle - \langle u_2 - z_2, Q(u_2 - u_1)\rangle \geq 0$$
$$\Rightarrow \langle z_2 - z_1, Q(u_2 - u_1)\rangle \geq \langle u_2 - u_1, Q(u_2 - u_1)\rangle \tag{18}$$

Also, observing the following

$$\langle (u_2 - u_1) - (z_2 - z_1), Q((u_2 - u_1) - (z_2 - z_1))\rangle \geq 0$$
$$\Rightarrow \langle u_2 - u_1, Q(z_2 - z_1)\rangle \leq \frac{1}{2}[\langle u_2 - u_1, Q(u_2 - u_1)\rangle + \langle z_2 - z_1, Q(z_2 - z_1)\rangle] \tag{19}$$

Combining (18) and (19), we have the required result. $\qquad\square$

## C.2 CONVERGENCE ANALYSIS OF ADMETAR FOR CONVEX OPTIMIZATION

**Lemma C.8.** *consider*

$$m_t = \beta_1 m_{t-1} + (1 - \beta_1)h_t, \ \forall t \geq 1.$$

*it follows that*

$$\langle h_t, \theta_t - \theta\rangle = \langle m_{t-1}, \theta_{t-1} - \theta\rangle$$
$$- \frac{\beta_1}{1 - \beta_1}\langle m_{t-1}, \theta_t - \theta_{t-1}\rangle$$
$$+ \frac{1}{1 - \beta_1}\left(\langle m_t, \theta_t - \theta\rangle - \langle m_{t-1}, \theta_{t-1} - \theta\rangle\right).$$

*Proof.* By definition of $m_t$, $h_t = \frac{1}{1-\beta_1}m_t - \frac{\beta_1}{1-\beta_1}m_{t-1}$. Thus, we have

$$\langle h_t, \theta_t - \theta\rangle = \frac{1}{1 - \beta_1}\langle m_t, \theta_t - \theta\rangle - \frac{\beta_1}{1 - \beta_1}\langle m_{t-1}, \theta_t - \theta\rangle$$
$$= \frac{1}{1 - \beta_1}\langle m_t, \theta_t - \theta\rangle - \frac{\beta_1}{1 - \beta_1}\langle m_{t-1}, \theta_{t-1} - \theta\rangle - \frac{\beta_1}{1 - \beta_1}\langle m_{t-1}, \theta_t - \theta_{t-1}\rangle$$
$$= \frac{1}{1 - \beta_1}\left(\langle m_t, \theta_t - \theta\rangle - \langle m_{t-1}, \theta_{t-1} - \theta\rangle\right) + \langle m_{t-1}, \theta_{t-1} - \theta\rangle$$
$$- \frac{\beta_1}{1 - \beta_1}\langle m_{t-1}, \theta_t - \theta_{t-1}\rangle.$$

$\qquad\square$

**Lemma C.9** (Bound for $\sum_{t=1}^T \alpha_t\|\hat{v}_t^{-1/4}m_t\|^2$). *Under Assumption in Theorem 1, we have*

$$\sum_{t=1}^T \alpha_t\|\hat{v}_t^{-1/4}m_t\|^2 \leq \frac{(1 - \beta_1)\alpha\sqrt{1 + \log T}}{\sqrt{(1 - \beta_2)(1 - \gamma)}} \sum_{i=1}^d \|h_{1:T,i}\|_2$$

*Proof.* First, we bound $\|\hat{v}_t^{-1/4}m_t\|^2$. From the definition of $m_t$ and $v_t$, it follows that

$$m_t = (1 - \beta_1)\sum_{j=1}^t \beta_1^{t-j}h_j, \ v_t = (1 - \beta_2)\sum_{j=1}^t \beta_2^{t-j}h_j^2$$

Then we have

$$
\|\hat{v}_t^{-1/4} m_t\|^2 \le \|v_t^{-1/4} m_t\|^2 = \sum_{i=1}^d \frac{m_{t,i}^2}{v_{t,i}^{1/2}} = \sum_{i=1}^d \frac{\left(\sum_{j=1}^t (1-\beta_1)\beta_1^{t-j} h_{j,i}\right)^2}{\sqrt{\sum_{j=1}^t (1-\beta_2)\beta_2^{t-j} h_{j,i}^2}}
$$

$$
= \frac{(1-\beta_1)^2}{\sqrt{1-\beta_2}} \sum_{i=1}^d \frac{\left(\sum_{j=1}^t \beta_1^{t-j} h_{j,i}\right)^2}{\sqrt{\sum_{j=1}^t \beta_2^{t-j} h_{j,i}^2}}
$$

$$
\le \frac{(1-\beta_1)^2}{\sqrt{1-\beta_2}} \Bigg(
$$

$$
\sum_{i=1}^d \frac{\left[\left(\sum_{j=1}^t (\beta_2^{\frac{t-j}{4}} |h_{j,i}|^{\frac{1}{2}})^4\right)^{\frac{1}{4}} \left(\sum_{j=1}^t (\beta_1^{1/2}\beta_2^{-1/4})^{4(t-j)}\right)^{\frac{1}{4}} \left(\sum_{j=1}^t (\beta_1^{t-j}|h_{j,i}|)^{\frac{1}{2}\cdot 2}\right)^{\frac{1}{2}}\right]^2}{\sqrt{\sum_{j=1}^t \beta_2^{t-j} h_{j,i}^2}} \Bigg)
$$

$$
= \frac{(1-\beta_1)^2}{\sqrt{1-\beta_2}} \sum_{i=1}^d \left(\sum_{j=1}^t \gamma^{t-j}\right)^{\frac{1}{2}} \sum_{j=1}^t \beta_1^{t-j}|h_{j,i}|
$$

$$
\le \frac{(1-\beta_1)^2}{\sqrt{(1-\beta_2)(1-\gamma)}} \sum_{i=1}^d \sum_{j=1}^t \beta_1^{t-j}|h_{j,i}|, \tag{20}
$$

where the first inequality follows from the fact that $\hat{v}_{t,i}^{1/2} \ge v_{t,i}^{1/2}$, the second one follows from the generalized Hölder inequality for

$$
\theta_j = \beta_2^{\frac{t-j}{4}} |h_{j,i}|^{\frac{1}{2}}, \quad y_j = (\beta_1 \beta_2^{-1/2})^{\frac{t-j}{2}}, \quad z_j = (\beta_1^{t-j}|h_{j,i}|)^{\frac{1}{2}} \quad \text{and} \quad p = q = 4, \quad r = 2,
$$

and the third one follows from the sum of geometric series and the assumption $\gamma = \frac{\beta_1^2}{\beta_2} < 1$. In this way, we can bound $\sum_{t=1}^T \alpha_t \|\hat{v}_t^{-1/4} m_t\|^2$.

$$
\sum_{t=1}^T \alpha_t \|\hat{v}_t^{-1/4} m_t\|^2 \le \frac{(1-\beta_1)^2}{\sqrt{(1-\beta_2)(1-\gamma)}} \sum_{i=1}^d \sum_{t=1}^T \alpha_t \sum_{j=1}^t \beta_1^{t-j}|h_{j,i}|
$$

$$
= \frac{(1-\beta_1)^2}{\sqrt{(1-\beta_2)(1-\gamma)}} \sum_{i=1}^d \sum_{j=1}^T \sum_{t=j}^T \alpha_t \beta_1^{t-j}|h_{j,i}|
$$

$$
\le \frac{(1-\beta_1)}{\sqrt{(1-\beta_2)(1-\gamma)}} \sum_{i=1}^d \sum_{j=1}^T \alpha_j |h_{j,i}|
$$

$$
\le \frac{1-\beta_1}{\sqrt{(1-\beta_2)(1-\gamma)}} \sum_{i=1}^d \sqrt{\sum_{j=1}^T \alpha_j^2} \sqrt{\sum_{j=1}^T h_{j,i}^2}
$$

$$
\le \frac{(1-\beta_1)\alpha\sqrt{1+\log T}}{\sqrt{(1-\beta_2)(1-\gamma)}} \sum_{i=1}^d \sqrt{\sum_{t=1}^T h_{t,i}^2}
$$

$$
= \frac{(1-\beta_1)\alpha\sqrt{1+\log T}}{\sqrt{(1-\beta_2)(1-\gamma)}} \sum_{i=1}^d \|h_{1:T,i}\|
$$

where the first inequality follows from (20). The first equality is by changing order of summation. The second inequality follows from the fact that $\sum_{t=j}^T \alpha_t \beta_1^{t-j} \le \frac{\alpha_j}{1-\beta_1}$. The third inequality is by Cauthy-Schwartz. The last inequality is by using $\sum_{j=1}^T \frac{1}{j} \le 1 + \log T$ □

**Theorem C.10.** *(Convergence of* ADMETAR *for convex optimization) Let $\{\theta_t\}$ be the sequence obtained from* ADMETAR, $0 \le \lambda, \beta_1, \beta_2 < 1$, $\gamma = \frac{\beta_1^2}{\beta_2} < 1$, $\alpha_t = \frac{\alpha}{\sqrt{t}}$ *and*

$v_t \leq v_{t+1}, \forall t \in [T]$. *Suppose* $x \in \mathcal{F}$, *where* $\mathcal{F} \subset \mathbb{R}^d$ *and has bounded diameter* $D_\infty$, *i.e.* $||\theta_t - \theta||_\infty \leq D_\infty, \forall t \in [T]$. *Assume* $f(\theta)$ *is a convex function and* $||g_t||_\infty$ *is bounded. Denote the optimal point as* $\theta$. *For* $\theta_t$ *generated,* ADMETAR *achieves the regret:*

$$R(T) = \sum_{t=1}^{T} [f_t(\theta_t) - f_t(\theta)] = \mathcal{O}(\sqrt{T})$$

*Proof.* • *Bound for* $\sum_{t=1}^{T} \langle m_t, \theta_t - \theta \rangle$.
As $x \in \mathcal{F}$, we get

$$\theta_{t+1} = \Pi_{\mathcal{F}, \sqrt{\hat{v}_t}}(\theta_t - \alpha_t \hat{v}_t^{-1/2} m_t) = \min_{x \in \mathcal{F}} \|\hat{v}_t^{1/4}(x - (\theta_t - \alpha_t \hat{v}_t^{-1/2} m_t))\|.$$

Furthermore, $\Pi_{\mathcal{F}, \sqrt{\hat{v}_t}}(x) = x$ for all $x \in \mathcal{F}$. Using Lemma C.7 with $u_1 = \theta_{t+1}$ and $u_2 = \theta$, we have the following:

$$\|\hat{v}_t^{1/4}(\theta_{t+1} - \theta)\|^2 \leq \|\hat{v}_t^{1/4}(\theta_t - \alpha_t \hat{v}_t^{-1/2} m_t - \theta)\|^2$$
$$= \|\hat{v}_t^{1/4}(\theta_t - \theta)\|^2 + \alpha_t^2 \|\hat{v}_t^{-1/4} m_t\|^2 - 2\alpha_t \langle m_t, \theta_t - \theta \rangle \quad (21)$$

we rearrange and divide both sides of (21) by $2\alpha_t$ to get

$$\langle m_t, \theta_t - \theta \rangle \leq \frac{1}{2\alpha_t} \|\hat{v}_t^{1/4}(\theta_t - \theta)\|^2 - \frac{1}{2\alpha_t} \|\hat{v}_t^{1/4}(\theta_{t+1} - \theta)\|^2 + \frac{\alpha_t}{2} \|\hat{v}_t^{-1/4} m_t\|^2$$
$$= \frac{1}{2\alpha_{t-1}} \|\hat{v}_{t-1}^{1/4}(\theta_t - \theta)\|^2 - \frac{1}{2\alpha_t} \|\hat{v}_t^{1/4}(\theta_{t+1} - \theta)\|^2$$
$$+ \frac{1}{2} \sum_{i=1}^{d} \left( \frac{\hat{v}_{t,i}^{1/2}}{\alpha_t} - \frac{\hat{v}_{t-1,i}^{1/2}}{\alpha_{t-1}} \right) (\theta_{t,i} - \theta_i)^2 + \frac{\alpha_t}{2} \|\hat{v}_t^{-1/4} m_t\|^2$$
$$\leq \frac{1}{2\alpha_{t-1}} \|\hat{v}_{t-1}^{1/4}(\theta_t - \theta)\|^2 - \frac{1}{2\alpha_t} \|\hat{v}_t^{1/4}(\theta_{t+1} - \theta)\|^2$$
$$+ \frac{D_\infty^2}{2} \sum_{i=1}^{d} \left( \frac{\hat{v}_{t,i}^{1/2}}{\alpha_t} - \frac{\hat{v}_{t-1,i}^{1/2}}{\alpha_{t-1}} \right) + \frac{\alpha_t}{2} \|\hat{v}_t^{-1/4} m_t\|^2 \quad (22)$$

where the last inequality is due to the fact that $\hat{v}_{t,i} \geq \hat{v}_{t-1,i}, \frac{1}{\alpha_t} \geq \frac{1}{\alpha_{t-1}}$, and the definition of $D_\infty$.

Summing (22) over $t = 1, \ldots T$ and using that $\hat{v}_0 = 0$ yields

$$\sum_{t=1}^{T} \langle m_t, \theta_t - \theta \rangle \leq \frac{D_\infty^2}{2\alpha_T} \sum_{i=1}^{d} \hat{v}_{T,i}^{1/2} + \frac{1}{2} \sum_{t=1}^{T} \alpha_t \|\hat{v}_t^{-1/4} m_t\|^2.$$

• *Bound for* $\sum_{t=1}^{T} \langle m_{t-1}, \theta_{t-1} - \theta_t \rangle$.

$$\sum_{t=1}^{T} \langle m_{t-1}, \theta_{t-1} - \theta_t \rangle = \sum_{t=2}^{T} \langle m_{t-1}, \theta_{t-1} - \theta_t \rangle = \sum_{t=1}^{T-1} \langle m_t, \theta_t - \theta_{t+1} \rangle$$
$$\leq \sum_{t=1}^{T-1} \|\hat{v}_t^{-1/4} m_t\| \|\hat{v}_t^{1/4}(\theta_{t+1} - \theta)\|$$
$$= \sum_{t=1}^{T-1} \|\hat{v}_t^{-1/4} m_t\| \left\| \hat{v}_t^{1/4} [\Pi_{\mathcal{F}, \hat{v}_t^{1/2}} \left( \theta_t - \alpha_t \hat{v}_t^{-1/2} m_t \right) - \Pi_{\mathcal{F}, \hat{v}_t^{1/2}}(\theta_t)] \right\|$$
$$\leq \sum_{t=1}^{T-1} \alpha_t \|\hat{v}_t^{-1/4} m_t\| \|\hat{v}_t^{-1/4} m_t\|$$
$$= \sum_{t=1}^{T-1} \alpha_t \|\hat{v}_t^{-1/4} m_t\|^2$$

where the first inequality follows from Hölder inequality and the second inequality is due to lemma C.7

- *Bound for $\langle m_T, \theta_T - \theta \rangle$.*

$$\langle m_T, \theta_T - \theta \rangle \leq \|\hat{v}_t^{-1/4} m_T\| \|\hat{v}_t^{1/4}(\theta_T - \theta)\|$$

$$\leq \alpha_T \|\hat{v}_t^{-1/4} m_T\|^2 + \frac{1}{4\alpha_T} \|\hat{v}_t^{1/4}(\theta_T - \theta)\|^2$$

$$\leq \alpha_T \|\hat{v}_t^{-1/4} m_T\|^2 + \frac{D_\infty^2}{4\alpha_T} \sum_{i=1}^d \hat{v}_{T,i}^{1/2}$$

where the first inequality follows from Hölder inequality and the second inequality follows from Young's inequality. The last inequality is due to the definition of $D_\infty$.

After all these preparations, we obtain:

$$\sum_{t=1}^T \langle h_t, \theta_t - \theta \rangle = \frac{\beta_1}{1-\beta_1} \left( \langle m_T, \theta_T - \theta \rangle + \sum_{t=1}^T \langle m_{t-1}, \theta_{t-1} - \theta_t \rangle \right) + \sum_{t=1}^T \langle m_t, \theta_t - \theta \rangle$$

$$\leq \frac{\beta_1}{1-\beta_1} \left( \frac{D_\infty^2}{4\alpha_T} \sum_{i=1}^d \hat{v}_{T,i}^{1/2} + \sum_{t=1}^T \alpha_t \|\hat{v}_t^{-1/4} m_t\|^2 \right) + \frac{D_\infty^2}{2\alpha_T} \sum_{i=1}^d \hat{v}_{T,i}^{1/2}$$

$$+ \frac{1}{2} \sum_{t=1}^T \alpha_t \|\hat{v}_t^{-1/4} m_t\|^2$$

$$= \frac{(2-\beta_1)D_\infty^2}{4\alpha_T(1-\beta_1)} \sum_{i=1}^d \hat{v}_{T,i}^{1/2} + \frac{2+\beta_1}{2(1-\beta_1)} \sum_{t=1}^T \alpha_t \|\hat{v}_t^{-1/4} m_t\|^2$$

$$\leq \frac{(2-\beta_1)D_\infty^2 \sqrt{T}}{4\alpha(1-\beta_1)} \sum_{i=1}^d \hat{v}_{T,i}^{1/2} + \frac{(2+\beta_1)\alpha\sqrt{1+\log T}}{2\sqrt{(1-\beta_2)(1-\gamma)}} \sum_{i=1}^d \|h_{1:T,i}\|_2$$

This proves that $\sum_{t=1}^T \langle h_t, \theta_t - \theta \rangle = \mathcal{O}(\sqrt{T})$. Suppose the optimizer runs for a long time, the bias of EMA is small (Zhuang et al., 2020), thus $E(I_t)$ approaches $E(g_t)$ as step increases. Since $h_t = \kappa g_t + \mu I_t$, $h_t$ is the same order as $g_t$ when the time is long enough, thus we have

$$\sum_{t=1}^T \langle g_t, \theta_t - \theta \rangle = \mathcal{O}(\sqrt{T}) \tag{23}$$

In addition, due to the convexity of $f(.)$, we have

$$R(T) = \sum_{t=1}^T f_t(\theta_t) - f_t(x) \leq \sum_{t=1}^T \langle g_t, \theta_t - \theta \rangle$$

Combined with (23), we complete the proof. □

### C.3 CONVERGENCE ANALYSIS OF ADMETAR FOR NON-CONVEX OPTIMIZATION

**Lemma C.11.** *Set $\theta_0 \triangleq x_1$ in Algorithm, and define $z_t$ as*

$$z_t = \theta_t + \frac{\beta_1}{1-\beta_1}(\theta_t - \theta_{t-1}), \ \forall t \geq 1. \tag{24}$$

*Then the following holds true*

$$z_{t+1} - z_t = -\frac{\beta_1}{1-\beta_1}\left(\frac{\alpha_t}{\sqrt{\hat{v}_t}} - \frac{\alpha_{t-1}}{\sqrt{\hat{v}_{t-1}}}\right)m_{t-1} - \alpha_t h_t / \sqrt{\hat{v}_t}$$

*Proof.* By the update of ADMETAR, we have

$$
\begin{aligned}
\theta_{t+1} - \theta_t &= -\alpha_t m_t / \sqrt{\hat{v}_t} = -\alpha_t (\beta_1 m_{t-1} + (1 - \beta_1) h_t) / \sqrt{\hat{v}_t} \\
&= \beta_1 \frac{\alpha_t}{\alpha_{t-1}} \frac{\sqrt{\hat{v}_{t-1}}}{\sqrt{\hat{v}_t}} (\theta_t - \theta_{t-1}) - \alpha_t (1 - \beta_1) h_t / \sqrt{\hat{v}_t} \\
&= \beta_1 (\theta_t - \theta_{t-1}) + \beta_1 \left( \frac{\alpha_t}{\alpha_{t-1}} \frac{\sqrt{\hat{v}_{t-1}}}{\sqrt{\hat{v}_t}} - 1 \right) (\theta_t - \theta_{t-1}) - \alpha_t (1 - \beta_1) h_t / \sqrt{\hat{v}_t} \\
&= \beta_1 (\theta_t - \theta_{t-1}) - \beta_1 \left( \frac{\alpha_t}{\sqrt{\hat{v}_t}} - \frac{\alpha_{t-1}}{\sqrt{\hat{v}_{t-1}}} \right) m_{t-1} - \alpha_t (1 - \beta_1) h_t / \sqrt{\hat{v}_t} \quad (25)
\end{aligned}
$$

Since we also have

$$
\theta_{t+1} - \theta_t = (1 - \beta_1)\theta_{t+1} + \beta_1(\theta_{t+1} - \theta_t) - (1 - \beta_1)\theta_t
$$

Combined with (25), we have

$$
\begin{aligned}
&(1 - \beta_1)\theta_{t+1} + \beta_1(\theta_{t+1} - \theta_t) \\
&= (1 - \beta_1)\theta_t + \beta_1(\theta_t - \theta_{t-1}) - \beta_1 \left( \frac{\alpha_t}{\sqrt{\hat{v}_t}} - \frac{\alpha_{t-1}}{\sqrt{\hat{v}_{t-1}}} \right) m_{t-1} - \alpha_t (1 - \beta_1) h_t / \sqrt{\hat{v}_t}.
\end{aligned}
$$

Divide both sides by $1 - \beta_1$, we have

$$
\begin{aligned}
&\theta_{t+1} + \frac{\beta_1}{1 - \beta_1}(\theta_{t+1} - \theta_t) \\
&= \theta_t + \frac{\beta_1}{1 - \beta_1}(\theta_t - \theta_{t-1}) - \frac{\beta_1}{1 - \beta_1} \left( \frac{\alpha_t}{\sqrt{\hat{v}_t}} - \frac{\alpha_{t-1}}{\sqrt{\hat{v}_{t-1}}} \right) m_{t-1} - \alpha_t h_t / \sqrt{\hat{v}_t}.
\end{aligned}
$$

$\square$

**Lemma C.12.** *Suppose that the conditions in Theorem C.2 hold, then*

$$
E\left[ f(z_{t+1}) - f(z_1) \right] \le \sum_{i=1}^{4} T_i, \quad (26)
$$

*where*

$$
\begin{aligned}
T_1 &= -E\left[ \sum_{i=1}^{t} \langle \nabla f(z_i), \frac{\beta_1}{1 - \beta_1} \left( \frac{\alpha_i}{\sqrt{\hat{v}_i}} - \frac{\alpha_{i-1}}{\sqrt{\hat{v}_{i-1}}} \right) m_{i-1} \rangle \right] \\
T_2 &= -E\left[ \sum_{i=1}^{t} \alpha_i \langle \nabla f(z_i), h_i / \sqrt{\hat{v}_i} \rangle \right] \\
T_3 &= E\left[ \sum_{i=1}^{t} L \left\| \frac{\beta_1}{1 - \beta_1} \left( \frac{\alpha_t}{\sqrt{\hat{v}_i}} - \frac{\alpha_{i-1}}{\sqrt{\hat{v}_{i-1}}} \right) m_{i-1} \right\|^2 \right] \\
T_4 &= E\left[ \sum_{i=1}^{t} L \left\| \alpha_i h_i / \sqrt{\hat{v}_i} \right\|^2 \right]
\end{aligned}
$$

*Proof.* By the Lipschitz smoothness of $\nabla f$,

$$
f(z_{t+1}) \le f(z_t) + \langle \nabla f(z_t), z_{t+1} - z_t \rangle + \frac{L}{2} \| z_{t+1} - z_t \|^2,
$$

Based on (C.18), we have

$$
\begin{aligned}
E[f(z_{t+1}) - f(z_1)] =& E\left[\sum_{i=1}^{t} f(z_{i+1}) - f(z_i)\right] \\
\leq& E\left[\sum_{i=1}^{t} \langle \nabla f(z_i), z_{i+1} - z_i \rangle + \frac{L}{2}\|z_{i+1} - z_i\|^2\right] \\
=& -E\left[\sum_{i=1}^{t} \langle \nabla f(z_i), \frac{\beta_1}{1-\beta_1}\left(\frac{\alpha_i}{\sqrt{\hat{v}_i}} - \frac{\alpha_{i-1}}{\sqrt{\hat{v}_{i-1}}}\right) m_{i-1} \rangle\right] \\
& - E\left[\sum_{i=1}^{t} \alpha_i \langle \nabla f(z_i), h_i/\sqrt{\hat{v}_i} \rangle\right] \\
& + E\left[\sum_{i=1}^{t} \frac{L}{2}\|z_{i+1} - z_i\|^2\right] = T_1 + T_2 + E\left[\sum_{i=1}^{t} \frac{L}{2}\|z_{i+1} - z_i\|^2\right],
\end{aligned}
$$

Then, using inequality $\|a + b\|^2 \leq 2\|a\|^2 + 2\|b\|^2$ and combined with lemma C.11,

$$
E\left[\sum_{i=1}^{t} \frac{L}{2}\|z_{i+1} - z_i\|^2\right] \leq T_3 + T_4
$$

$\square$

**Lemma C.13.** *In this part, we bound $T_1, T_2, T_3$*

*Proof.* • *Bound for $T_1$*

$$
\begin{aligned}
T_1 =& -E\left[\sum_{i=2}^{t} \langle \nabla f(z_i), \frac{\beta_1}{1-\beta_1}\left(\frac{\alpha_i}{\sqrt{\hat{v}_i}} - \frac{\alpha_{i-1}}{\sqrt{\hat{v}_{i-1}}}\right) m_{i-1} \rangle\right] \\
\leq& E\left[\sum_{i=1}^{t} \|\nabla f(z_i)\| \|m_{i-1}\| \left(\frac{1}{1-\beta_1} - 1\right) \sum_{j=1}^{d} \left|\left(\frac{\alpha_i}{\sqrt{\hat{v}_i}} - \frac{\alpha_{i-1}}{\sqrt{\hat{v}_{i-1}}}\right)_j\right|\right] \\
\leq& H^2 \frac{\beta_1}{1-\beta_1} E\left[\sum_{i=1}^{t} \sum_{j=1}^{d} \left|\left(\frac{\alpha_i}{\sqrt{\hat{v}_i}} - \frac{\alpha_{i-1}}{\sqrt{\hat{v}_{i-1}}}\right)_j\right|\right]
\end{aligned}
$$

• *Bound for $T_3$*

$$
\begin{aligned}
T_3 \leq& LE\left[\sum_{i=2}^{t} \left(\frac{\beta_1}{1-\beta_1}\right)^2 \sum_{j=1}^{d} \left(\left(\frac{\alpha_t}{\sqrt{\hat{v}_i}} - \frac{\alpha_{i-1}}{\sqrt{\hat{v}_{i-1}}}\right)_j^2 (m_{i-1})_j^2\right)\right] \\
\leq& \left(\frac{\beta_1}{1-\beta_1}\right)^2 LH^2 E\left[\sum_{i=2}^{t} \sum_{j=1}^{d} \left(\frac{\alpha_t}{\sqrt{\hat{v}_i}} - \frac{\alpha_{i-1}}{\sqrt{\hat{v}_{i-1}}}\right)_j^2\right]
\end{aligned}
$$

• *Bound for $T_2$*

$$
\begin{aligned}
T_2 =& -E\left[\sum_{i=1}^{t} \alpha_i \langle \nabla f(z_i), h_i/\sqrt{\hat{v}_i} \rangle\right] \\
=& -E\left[\sum_{i=1}^{t} \alpha_i \langle \nabla f(\theta_i), h_i/\sqrt{\hat{v}_i} \rangle\right] - E\left[\sum_{i=1}^{t} \alpha_i \langle \nabla f(z_i) - \nabla f(\theta_i), h_i/\sqrt{\hat{v}_i} \rangle\right]. \quad (27)
\end{aligned}
$$

The second term of (27) can be bounded as

$$
-E\left[\sum_{i=1}^{t}\alpha_i\langle\nabla f(z_i)-\nabla f(\theta_i), h_i/\sqrt{\hat{v}_i}\rangle\right]
$$

$$
\leq E\left[\sum_{i=2}^{t}\frac{1}{2}\|\nabla f(z_i)-\nabla f(\theta_i)\|^2+\frac{1}{2}\|\alpha_i h_i/\sqrt{\hat{v}_i}\|^2\right]
$$

$$
\leq\frac{L^2}{2}E\left[\sum_{i=2}^{t}\left\|\frac{\beta_1}{1-\beta_1}\alpha_{i-1}m_{i-1}/\sqrt{\hat{v}_{i-1}}\right\|^2\right]+\frac{1}{2}E\left[\sum_{i=2}^{t}\|\alpha_i h_i/\sqrt{\hat{v}_i}\|^2\right]
$$

$$
=\frac{L^2}{2}\left(\frac{\beta_1}{1-\beta_1}\right)^2 E\left[\sum_{i=2}^{t}\left\|\alpha_{i-1}m_{i-1}/\sqrt{\hat{v}_{i-1}}\right\|^2\right]+\frac{1}{2}E\left[\sum_{i=2}^{t}\|\alpha_i h_i/\sqrt{\hat{v}_i}\|^2\right]
$$

where the second inequality is due to $\|\nabla f(z_i)-\nabla f(\theta_i)\|\leq L\|z_i-\theta_i\|$.
Then consider the first term of (27)

$$
E\left[\sum_{i=1}^{t}\alpha_i\langle\nabla f(\theta_i), h_i/\sqrt{\hat{v}_i}\rangle\right]
$$

$$
=\kappa E\left[\sum_{i=1}^{t}\alpha_i\langle\nabla f(\theta_i), g_i/\sqrt{\hat{v}_i}\rangle\right]+\mu E\left[\sum_{i=1}^{t}\alpha_i\langle\nabla f(\theta_i), I_i/\sqrt{\hat{v}_i}\rangle\right]
$$

Consider the term with $\kappa$

$$
E\left[\sum_{i=1}^{t}\alpha_i\langle\nabla f(\theta_i), g_i/\sqrt{\hat{v}_i}\rangle\right]
$$

$$
=E\left[\sum_{i=1}^{t}\alpha_i\langle\nabla f(\theta_i), (\nabla f(\theta_i)+\delta_i)/\sqrt{\hat{v}_i}\rangle\right]
$$

$$
=E\left[\sum_{i=1}^{t}\alpha_i\langle\nabla f(\theta_i), \nabla f(\theta_i)/\sqrt{\hat{v}_i}\rangle\right]+E\left[\sum_{i=1}^{t}\alpha_i\langle\nabla f(\theta_i), \delta_i/\sqrt{\hat{v}_i}\rangle\right]. \tag{28}
$$

For the second term in RHS of (28), we have

$$
E\left[\sum_{i=1}^{t}\alpha_i\langle\nabla f(\theta_i), \delta_i/\sqrt{\hat{v}_i}\rangle\right]
$$

$$
=E\left[\sum_{i=2}^{t}\langle\nabla f(\theta_i), \delta_i(\alpha_i/\sqrt{\hat{v}_i}-\alpha_{i-1}/\sqrt{\hat{v}_{i-1}})\rangle\right]+E\left[\sum_{i=2}^{t}\alpha_{i-1}\langle\nabla f(\theta_i), \delta_i(1/\sqrt{\hat{v}_{i-1}})\rangle\right]
$$

$$
+E\left[\alpha_1\langle\nabla f(x_1), \delta_1/\sqrt{\hat{v}_1}\rangle\right]
$$

$$
\geq E\left[\sum_{i=2}^{t}\langle\nabla f(\theta_i), \delta_i(\alpha_i/\sqrt{\hat{v}_i}-\alpha_{i-1}/\sqrt{\hat{v}_{i-1}})\rangle\right]-2H^2 E\left[\sum_{j=1}^{d}(\alpha_1/\sqrt{\hat{v}_1})_j\right] \tag{29}
$$

where the last equation is because given $\theta_i, \hat{v}_{i-1}$, $E\left[\delta_i(1/\sqrt{\hat{v}_{i-1}})|\theta_i, \hat{v}_{i-1}\right] = 0$ and $\|\delta_i\| \le 2H$
Further, we have

$$E\left[\sum_{i=2}^{t}\langle\nabla f(\theta_i), \delta_t(\alpha_i/\sqrt{\hat{v}_i} - \alpha_{i-1}/\sqrt{\hat{v}_{i-1}})\rangle\right]$$

$$=E\left[\sum_{i=2}^{t}\sum_{j=1}^{d}(\nabla f(\theta_i))_j(\delta_t)_j(\alpha_i/(\sqrt{\hat{v}_i})_j - \alpha_{i-1}/(\sqrt{\hat{v}_{i-1}})_j)\right]$$

$$\ge -E\left[\sum_{i=2}^{t}\sum_{j=1}^{d}|(\nabla f(\theta_i))_j|\,|(\delta_t)_j|\,\left|(\alpha_i/(\sqrt{\hat{v}_i})_j - \alpha_{i-1}/(\sqrt{\hat{v}_{i-1}})_j)\right|\right]$$

$$\ge -2H^2 E\left[\sum_{i=2}^{t}\sum_{j=1}^{d}\left|(\alpha_i/(\sqrt{\hat{v}_i})_j - \alpha_{i-1}/(\sqrt{\hat{v}_{i-1}})_j)\right|\right] \tag{30}$$

Substitute (29) and (30) into (28), we then get

$$-E\left[\sum_{i=1}^{t}\alpha_i\langle\nabla f(\theta_i), g_i/\sqrt{\hat{v}_i}\rangle\right]$$

$$\le 2H^2 E\left[\sum_{i=2}^{t}\sum_{j=1}^{d}\left|(\alpha_i/(\sqrt{\hat{v}_i})_j - \alpha_{i-1}/(\sqrt{\hat{v}_{i-1}})_j)\right|\right] + 2H^2 E\left[\sum_{j=1}^{d}(\alpha_1/\sqrt{\hat{v}_1})_j\right]$$

$$-E\left[\sum_{i=1}^{t}\alpha_i\langle\nabla f(\theta_i), \nabla f(\theta_i)/\sqrt{\hat{v}_i}\rangle\right] \tag{31}$$

Then, consider the term with $\mu$. Suppose the optimizer runs for a long time, the bias of EMA is small (Zhuang et al., 2020), thus $E(I_t)$ approaches $E(g_t)$ as step increases. In other words, we can bound it the same way as the term with $\kappa$.
After all these bounds, we finally get

$$T_2 \le \frac{L^2}{2}E\left[\sum_{i=2}^{t}\left\|\frac{\beta_1}{1-\beta_1}\alpha_{i-1}m_{i-1}/\sqrt{\hat{v}_{i-1}}\right\|^2\right] + \frac{1}{2}E\left[\sum_{i=2}^{t}\|\alpha_i h_i/\sqrt{\hat{v}_i}\|^2\right]$$

$$+ 2(\kappa+\mu)H^2 E\left[\sum_{i=2}^{t}\sum_{j=1}^{d}\left|(\alpha_i/(\sqrt{\hat{v}_i})_j - \alpha_{i-1}/(\sqrt{\hat{v}_{i-1}})_j)\right|\right]$$

$$+ 2(\kappa+\mu)H^2 E\left[\sum_{j=1}^{d}(\alpha_1/\sqrt{\hat{v}_1})_j\right] - (\kappa+\mu)E\left[\sum_{i=1}^{t}\alpha_i\langle\nabla f(\theta_i), \nabla f(\theta_i)/\sqrt{\hat{v}_i}\rangle\right]$$

$\square$

**Lemma C.14.** *Suppose the conditions in theorem C.2 holds, then we have*

$$E\left[\sum_{i=1}^{t}\alpha_i\langle\nabla f(\theta_i), \nabla f(\theta_i)/\sqrt{\hat{v}_i}\rangle\right]$$

$$\le E\left[C_1\sum_{i=1}^{t}\left\|\frac{\alpha_t h_t}{\sqrt{\hat{v}_t}}\right\|^2 + C_2\sum_{i=2}^{t}\left\|\frac{\alpha_{i-1}m_{i-1}}{\sqrt{\hat{v}_{i-1}}}\right\|^2 + C_3\sum_{i=2}^{t}\left\|\frac{\alpha_t}{\sqrt{\hat{v}_t}} - \frac{\alpha_{t-1}}{\sqrt{\hat{v}_{t-1}}}\right\|_1\right.$$

$$\left.+ C_4\sum_{i=2}^{t-1}\left\|\frac{\alpha_t}{\sqrt{\hat{v}_t}} - \frac{\alpha_{t-1}}{\sqrt{\hat{v}_{t-1}}}\right\|^2\right] + C_5$$

*where $C_1, C_2, C_3, C_4$ and $C_5$ are independent of the step.*

*Proof.* Combine lemma C.12 and lemma C.13, we get

$$E\left[f(z_{t+1}) - f(z_1)\right]$$

$$\leq H^2 \frac{\beta_1}{1-\beta_1} E\left[\sum_{i=1}^{t}\sum_{j=1}^{d}\left|\left(\frac{\alpha_i}{\sqrt{\hat{v}_i}} - \frac{\alpha_{i-1}}{\sqrt{\hat{v}_{i-1}}}\right)_j\right|\right]$$

$$+ \left(\frac{\beta_1}{1-\beta_1}\right)^2 LH^2 E\left[\sum_{i=2}^{t}\sum_{j=1}^{d}\left(\frac{\alpha_t}{\sqrt{\hat{v}_i}} - \frac{\alpha_{i-1}}{\sqrt{\hat{v}_{i-1}}}\right)_j^2\right]$$

$$+ E\left[\sum_{i=1}^{t} L \left\|\alpha_i h_i / \sqrt{\hat{v}_i}\right\|^2\right]$$

$$+ \frac{L^2}{2} E\left[\sum_{i=2}^{t}\left\|\frac{\beta_1}{1-\beta_1}\alpha_{i-1}m_{i-1}/\sqrt{\hat{v}_{i-1}}\right\|^2\right] + \frac{1}{2} E\left[\sum_{i=2}^{t}\|\alpha_i h_i/\sqrt{\hat{v}_i}\|^2\right]$$

$$+ 2(\kappa+\mu)H^2 E\left[\sum_{i=2}^{t}\sum_{j=1}^{d}\left|\left(\frac{\alpha_i}{\sqrt{\hat{v}_i}} - \frac{\alpha_{i-1}}{\sqrt{\hat{v}_{i-1}}}\right)_j\right|\right]$$

$$+ 2(\kappa+\mu)H^2 E\left[\sum_{j=1}^{d}(\alpha_1/\sqrt{\hat{v}_1})_j\right] - (\kappa+\mu)E\left[\sum_{i=1}^{t}\alpha_i\langle\nabla f(\theta_i), \nabla f(\theta_i)/\sqrt{\hat{v}_i}\rangle\right]$$

By merging similar terms in above inequality and noticing that $\kappa + \mu > 0$, we get

$$E\left[\sum_{i=1}^{t}\alpha_i\langle\nabla f(\theta_i), \nabla f(\theta_i)/\sqrt{\hat{v}_i}\rangle\right]$$

$$\leq \left(2H^2 + \frac{\beta_1 H^2}{(1-\beta_1)(\kappa+\mu)}\right) E\left[\sum_{i=1}^{t}\sum_{j=1}^{d}\left|\left(\frac{\alpha_i}{\sqrt{\hat{v}_i}} - \frac{\alpha_{i-1}}{\sqrt{\hat{v}_{i-1}}}\right)_j\right|\right]$$

$$+ \left(\frac{\beta_1}{1-\beta_1}\right)^2 \frac{LH^2}{\kappa+\mu} E\left[\sum_{i=2}^{t}\sum_{j=1}^{d}\left(\frac{\alpha_t}{\sqrt{\hat{v}_i}} - \frac{\alpha_{i-1}}{\sqrt{\hat{v}_{i-1}}}\right)_j^2\right]$$

$$+ \left(\frac{2L+1}{2(\kappa+\mu)}\right) E\left[\sum_{i=2}^{t}\|\frac{\alpha_i h_i}{\sqrt{\hat{v}_i}}\|^2\right] + \frac{L^2}{2(\kappa+\mu)}\left(\frac{\beta_1}{1-\beta_1}\right)^2 E\left[\sum_{i=2}^{t}\left\|\frac{\alpha_{i-1}m_{i-1}}{\sqrt{\hat{v}_{i-1}}}\right\|^2\right]$$

$$+ 2H^2 E\left[\sum_{j=1}^{d}(\alpha_1/\sqrt{\hat{v}_1})_j\right] + \frac{1}{\kappa+\mu}E\left[f(z_1) - f(z_{t+1})\right]$$

$$= E\left[C_1 \sum_{i=1}^{t}\left\|\frac{\alpha_t h_t}{\sqrt{\hat{v}_t}}\right\|^2 + C_2 \sum_{i=2}^{t}\left\|\frac{\alpha_{i-1}m_{i-1}}{\sqrt{\hat{v}_{i-1}}}\right\|^2 + C_3 \sum_{i=2}^{t}\left\|\frac{\alpha_t}{\sqrt{\hat{v}_t}} - \frac{\alpha_{t-1}}{\sqrt{\hat{v}_{t-1}}}\right\|_1\right.$$

$$\left. + C_4 \sum_{i=2}^{t-1}\left\|\frac{\alpha_t}{\sqrt{\hat{v}_t}} - \frac{\alpha_{t-1}}{\sqrt{\hat{v}_{t-1}}}\right\|^2\right] + C_5 \tag{32}$$

$\square$

**Theorem C.15.** *(Convergence of* ADMETAR *for non-convex optimization)*
*Under the assumptions:*

- $\nabla f$ *exits and is Lipschitz-continuous,i.e,* $||\nabla f(x) - \nabla f(y)|| \leq L||x - y||$, $\forall x, y$; $f$ *is also lower bounded.*

- *At step $t$, the algorithm can access a bounded noisy gradient $g_t$, and the true gradient $\nabla f$ is also bounded.*

- *The noisy gradient is unbiased, and has independent noise, i.e. $g_t = \nabla f(\theta_t) + \delta_t, \mathbb{E}[\delta_t] = 0$ and $\delta_i \perp \delta_j, \forall i \neq j$.*

*Assume $\min_{j \in [d]}(v_1)_j \geq c > 0$ and $\alpha_t = \alpha/\sqrt{t}$, then for any T we have:*

$$\min_{t \in [T]} \mathbb{E}\left\|\nabla f(\theta_t)\right\|^2 \leq \tfrac{1}{\sqrt{T}}(Q_1 + Q_2 \log T)$$

*where $Q_1$ and $Q_2$ are constants independent of T.*

*Proof.* We bound non-constant terms in RHS of (32), which is given by

$$E\left[C_1 \sum_{t=1}^{T}\left\|\frac{\alpha_t h_t}{\sqrt{\hat{v}_t}}\right\|^2 + C_2 \sum_{i=2}^{t}\left\|\frac{\alpha_{i-1} m_{i-1}}{\sqrt{\hat{v}_{i-1}}}\right\|^2 + C_3 \sum_{t=2}^{T}\left\|\frac{\alpha_t}{\sqrt{\hat{v}_t}} - \frac{\alpha_{t-1}}{\sqrt{\hat{v}_{t-1}}}\right\|_1 \right.$$
$$\left. + C_4 \sum_{t=2}^{T-1}\left\|\frac{\alpha_t}{\sqrt{\hat{v}_t}} - \frac{\alpha_{t-1}}{\sqrt{\hat{v}_{t-1}}}\right\|^2\right] + C_5$$

- *Bound the term with $C_1$.*
Note that $\min_{j \in [d]}(\sqrt{\hat{v}_1})_j \geq \min_{j \in [d]}|(h_1)_j| \geq c > 0$, thus we have

$$E\left[\sum_{t=1}^{T}\left\|\frac{\alpha_t h_t}{\sqrt{\hat{v}_t}}\right\|^2\right]$$
$$\leq E\left[\sum_{t=1}^{T}\left\|\frac{\alpha_t h_t}{c}\right\|^2\right] = E\left[\sum_{t=1}^{T}\left\|\frac{\alpha h_t}{c\sqrt{t}}\right\|^2\right] = E\left[\sum_{t=1}^{T}\left(\frac{\alpha}{c\sqrt{t}}\right)^2 \|h_t\|^2\right]$$
$$\leq \frac{H^2 \alpha^2}{c^2} \sum_{t=1}^{T} \frac{1}{t} \leq \frac{H^2 \alpha^2}{c^2}(1 + \log T)$$

where the first inequality is due to $(\hat{v}_t)_j \geq (\hat{v}_{t-1})_j$, and the last inequality is due to $\sum_{t=1}^{T} \frac{1}{t} \leq 1 + \log T$.
- *Bound the term with $C_2$.*
Apply the same proof as above, we get

$$\sum_{i=2}^{t}\left\|\frac{\alpha_{i-1} m_{i-1}}{\sqrt{\hat{v}_{i-1}}}\right\|^2 \leq \frac{H^2 \alpha^2}{c^2}(1 + \log T)$$

- *Bound the term with $C_3$.*

$$E\left[\sum_{t=2}^{T}\left\|\frac{\alpha_t}{\sqrt{\hat{v}_t}} - \frac{\alpha_{t-1}}{\sqrt{\hat{v}_{t-1}}}\right\|_1\right] = E\left[\sum_{j=1}^{d}\sum_{t=2}^{T}\left(\frac{\alpha_{t-1}}{(\sqrt{\hat{v}_{t-1}})_j} - \frac{\alpha_t}{(\sqrt{\hat{v}_t})_j}\right)\right]$$
$$= E\left[\sum_{j=1}^{d}\left(\frac{\alpha_1}{(\sqrt{\hat{v}_1})_j} - \frac{\alpha_T}{(\sqrt{\hat{v}_T})_j}\right)\right] \leq E\left[\sum_{j=1}^{d}\frac{\alpha_1}{(\sqrt{\hat{v}_1})_j}\right] \leq \frac{d\alpha}{c} \tag{33}$$

where the first equality is due to $(\hat{v}_t)_j \geq (\hat{v}_{t-1})_j$ and $\alpha_t \leq \alpha_{t-1}$, and the second equality is due to telescope sum.

• *Bound the term with $C_4$.*

$$
E\left[\sum_{t=2}^{T-1}\left\|\frac{\alpha_t}{\sqrt{\hat{v}_t}}-\frac{\alpha_{t-1}}{\sqrt{\hat{v}_{t-1}}}\right\|^2\right]
$$

$$
=E\left[\sum_{t=2}^{T-1}\sum_{j=1}^{d}\left(\frac{\alpha_t}{\sqrt{\hat{v}_t}}-\frac{\alpha_{t-1}}{\sqrt{\hat{v}_{t-1}}}\right)_i^2\right]
$$

$$
\leq E\left[\sum_{t=2}^{T-1}\sum_{j=1}^{d}\frac{\alpha}{c}\left|\frac{\alpha_t}{\sqrt{\hat{v}_t}}-\frac{\alpha_{t-1}}{\sqrt{\hat{v}_{t-1}}}\right|_i\right]
$$

$$
\leq\frac{d\alpha^2}{c^2}
$$

where the first inequality is due to $|(\alpha_t/\sqrt{\hat{v}_t}-\alpha_{t-1}/\sqrt{\hat{v}_{t-1}})_j|\leq 1/c$.

Then we have for ADMETAR,

$$
E\left[C_1\sum_{t=1}^{T}\left\|\frac{\alpha_t h_t}{\sqrt{\hat{v}_t}}\right\|^2+C_2\sum_{i=2}^{t}\left\|\frac{\alpha_{i-1}m_{i-1}}{\sqrt{\hat{v}_{i-1}}}\right\|^2+C_3\sum_{t=2}^{T}\left\|\frac{\alpha_t}{\sqrt{\hat{v}_t}}-\frac{\alpha_{t-1}}{\sqrt{\hat{v}_{t-1}}}\right\|_1\right. \tag{34}
$$

$$
\left.+C_4\sum_{t=2}^{T-1}\left\|\frac{\alpha_t}{\sqrt{\hat{v}_t}}-\frac{\alpha_{t-1}}{\sqrt{\hat{v}_{t-1}}}\right\|^2\right]+C_5 \tag{35}
$$

$$
\leq C_1\frac{H^2\alpha^2}{c^2}(1+\log T)+C_2\frac{H^2\alpha^2}{c^2}(1+\log T)+C_3\frac{d\alpha}{c}+C_4\frac{d\alpha^2}{c^2}+C_5 \tag{36}
$$

Furthermore, due to $\|g_t\|\leq H$, we have $(\hat{v}_t)_j\leq H^2$, then we get

$$
\alpha/(\sqrt{\hat{v}_t})_j\geq\frac{1}{H\sqrt{t}}
$$

Thus we have

$$
E\left[\sum_{t=1}^{T}\alpha_i\langle\nabla f(\theta_t),\nabla f(\theta_t)/\sqrt{\hat{v}_t}\rangle\right]\geq E\left[\sum_{t=1}^{T}\frac{1}{H\sqrt{t}}\|\nabla f(\theta_t)\|^2\right]\geq\frac{\sqrt{T}}{H}\min_{t\in[T]}E\left[\|\nabla f(\theta_t)\|^2\right] \tag{37}
$$

Combine (36) and (37), we have

$$
\min_{t\in[T]}E\left[\|\nabla f(\theta_t)\|^2\right]
$$

$$
\leq\frac{H}{\sqrt{T}}\left((C_1+C_2)\frac{H^2\alpha^2}{c^2}(1+\log T)+C_3\frac{d\alpha}{c}+C_4\frac{d\alpha^2}{c^2}+C_5\right)
$$

$$
=\frac{1}{\sqrt{T}}\left(Q_1+Q_2\log T\right)
$$

This completes the proof. □

## C.4 CONVERGENCE ANALYSIS OF ADMETAS FOR CONVEX OPTIMIZATION

**Lemma C.16** (Bound for $\sum_{t=1}^{T}\alpha_t\|m_t\|^2$). *Under Assumption in Theorem 3, we have*

$$
\sum_{t=1}^{T}\alpha_t\|m_t\|^2\leq 2\alpha dG_\infty^2\sqrt{T}
$$

*Proof.* First, we bound $\|m_t\|$.

$$\|m_t\|^2 \leq d\|m_t\|_\infty^2 \leq dG_\infty^2 \tag{38}$$

Now we can bound $\sum_{t=1}^{T} \alpha_t \|m_t\|^2$

$$\sum_{t=1}^{T} \alpha_t \|m_t\|^2 \leq dG_\infty^2 \sum_{t=1}^{T} \alpha_t = \alpha dG_\infty^2 \sum_{t=1}^{T} \frac{1}{\sqrt{t}} \leq 2\alpha dG_\infty^2 \sqrt{T}$$

$\square$

**Theorem C.17.** *(Convergence of* ADMETAS *for convex optimization)*
*Let* $\{\theta_t\}$ *be the sequence obtained by* ADMETAS, $0 \leq \lambda, \beta < 1$, $\alpha_t = \frac{\alpha}{\sqrt{t}}$,
$\forall t \in [T]$. *Suppose* $x \in \mathcal{F}$, *where* $\mathcal{F} \subset \mathbb{R}^d$ *and has bounded diameter* $D_\infty$, *i.e.*
$\|\theta_t - \theta\|_\infty \leq D_\infty, \forall t \in [T]$.. *Assume* $f(\theta)$ *is a convex function and* $\|g_t\|_\infty$ *is*
*bounded. Denote the optimal point as* $\theta$. *For* $\theta_t$ *generated,* ADMETAS *achieves the regret:*

$$R(T) = \sum_{t=1}^{T} [f_t(\theta_t) - f_t(\theta)] = \mathcal{O}(\sqrt{T})$$

*Proof.* • *Bound for* $\sum_{t=1}^{T} \langle m_t, \theta_t - \theta \rangle$.
From the update process, we get

$$\|\theta_{t+1} - \theta\|^2 = \|\theta_t - \theta - \alpha_t m_t\|^2 = \|\theta_t - \theta\|^2 - 2\alpha_t \langle m_t, \theta_t - \theta \rangle + \alpha_t^2 \|m_t\|^2$$

thus we have

$$\sum_{t=1}^{T} \langle m_t, \theta_t - \theta \rangle = \sum_{t=1}^{T} \frac{1}{2\alpha_t} \left( \|\theta_t - \theta\|^2 - \|\theta_{t+1} - \theta\|^2 \right) + \sum_{i=1}^{T} \frac{\alpha_t}{2} \|m_t\|^2$$

consider the left-hand side

$$\sum_{t=1}^{T} \frac{1}{2\alpha_t} \left( \|\theta_t - \theta\|^2 - \|\theta_{t+1} - \theta\|^2 \right)$$

$$= \frac{1}{2\alpha_1} \|\theta_1 - \theta\|^2 + \sum_{t=2}^{T} \left( \frac{1}{2\alpha_t} - \frac{1}{2\alpha_{t-1}} \right) \|\theta_t - \theta\|^2 - \frac{1}{2\alpha_T} \|\theta_{T+1} - \theta\|^2$$

$$\leq \frac{dD_\infty^2}{2\alpha_1} + dD_\infty^2 \sum_{t=2}^{T} \left( \frac{1}{2\alpha_t} - \frac{1}{2\alpha_{t-1}} \right) + 0 = \frac{dD_\infty^2}{2\alpha_T}$$

Finally,we get

$$\sum_{t=1}^{T} \langle m_t, \theta_t - \theta \rangle \leq \frac{dD_\infty^2}{2\alpha_T} + \sum_{i=1}^{T} \frac{\alpha_t}{2} \|m_t\|^2$$

• *Bound for* $\sum_{t=1}^{T} \langle m_{t-1}, \theta_{t-1} - \theta_t \rangle$.

$$\sum_{t=1}^{T} \langle m_{t-1}, \theta_{t-1} - \theta_t \rangle = \sum_{t=1}^{T-1} \langle m_t, \theta_t - \theta_{t+1} \rangle$$

$$= \sum_{t=1}^{T-1} \langle m_t, \alpha_t m_t \rangle$$

$$= \sum_{t=1}^{T-1} \alpha_t \|m_t\|^2$$

• *Bound for $\langle m_T, \theta_T - \theta \rangle$.*

$$\langle m_T, \theta_T - \theta \rangle \leq \alpha_T \|m_T\|^2 + \frac{1}{4\alpha_T} \|\theta_T - \theta\|^2$$

$$\leq \alpha_T \|m_T\|^2 + \frac{dD_\infty^2}{4\alpha_T}$$

where the first inequality follows from Young's inequality.

Combine all these preparations, we obtain

$$\sum_{t=1}^{T} \langle h_t, \theta_t - \theta \rangle = \frac{1}{1-\beta} \left( \langle m_T, \theta_T - \theta \rangle - \langle m_0, \theta_0 - \theta \rangle \right) + \langle m_0, \theta_0 - \theta \rangle$$

$$+ \sum_{t=1}^{T-1} \langle m_t, \theta_t - \theta \rangle + \frac{\beta}{1-\beta} \sum_{t=1}^{T} \langle m_{t-1}, \theta_{t-1} - \theta_t \rangle$$

$$= \frac{\beta}{1-\beta} \langle m_T, \theta_T - \theta \rangle + \frac{\beta}{1-\beta} \sum_{t=1}^{T} \langle m_{t-1}, \theta_{t-1} - \theta_t \rangle + \sum_{t=1}^{T} \langle m_t, \theta_t - \theta \rangle$$

$$\leq \frac{\beta}{1-\beta} \left( \frac{dD_\infty}{4\alpha_T} + \sum_{t=1}^{T} \alpha_t \|m_t\|^2 \right) + \frac{dD_\infty^2}{2\alpha_T} + \sum_{i=1}^{T} \frac{\alpha_t}{2} \|m_t\|^2$$

$$\leq \left( \frac{\beta}{1-\beta} + 2 \right) \frac{dD_\infty}{4\alpha_T} + \left( \frac{2\alpha\beta}{1-\beta} + \alpha \right) dG_\infty^2 \sqrt{T}$$

This proves that $\sum_{t=1}^{T} \langle h_t, \theta_t - \theta \rangle = \mathcal{O}(\sqrt{T})$. Suppose the optimizer runs for a long time, the bias of EMA is small (Zhuang et al., 2020), thus $E(I_t)$ approaches $E(g_t)$ as step increases. Since $h_t = \kappa g_t + \mu I_t$, $h_t$ is the same order as $g_t$ when the time is long enough, thus we have

$$\sum_{t=1}^{T} \langle g_t, \theta_t - \theta \rangle = \mathcal{O}(\sqrt{T}) \tag{39}$$

In addition, due to the convexity of $f(.)$, we have

$$R(T) = \sum_{t=1}^{T} f_t(\theta_t) - f_t(x) \leq \sum_{t=1}^{T} \langle g_t, \theta_t - \theta \rangle$$

Combined with (39), we complete the proof. □

## C.5 CONVERGENCE ANALYSIS OF ADMETAS FOR NON-CONVEX OPTIMIZATION

**Lemma C.18.** *Set $\theta_0 \triangleq \theta_1$ in Algorithm, and define $z_t$ as*

$$z_t = \theta_t + \frac{\beta}{1-\beta} (\theta_t - \theta_{t-1}), \ \forall t \geq 1. \tag{40}$$

*Then the following holds*

$$z_{t+1} - z_t = -\frac{\beta}{1-\beta} (\alpha_t - \alpha_{t-1}) m_{t-1} - \alpha_t h_t$$

*Proof.* By the update rule of ADMETAS, we have

$$\theta_{t+1} - \theta_t = -\alpha_t m_t = -\alpha_t [\beta m_{t-1} + (1-\beta) h_t]$$

$$= \beta \frac{\alpha_t}{\alpha_{t-1}} (\theta_t - \theta_{t-1}) - \alpha_t (1-\beta) h_t$$

$$= \beta (\theta_t - \theta_{t-1}) + \beta \left( \frac{\alpha_t}{\alpha_{t-1}} - 1 \right) (\theta_t - \theta_{t-1}) - \alpha_t (1-\beta) h_t$$

$$= \beta (\theta_t - \theta_{t-1}) - \beta (\alpha_t - \alpha_{t-1}) m_{t-1} - \alpha_t (1-\beta) h_t \tag{41}$$

Since we also have
$$\theta_{t+1} - \theta_t = (1-\beta)\theta_{t+1} + \beta(\theta_{t+1} - \theta_t) - (1-\beta)\theta_t$$

Combined with (41), we have
$$(1-\beta)\theta_{t+1} + \beta(\theta_{t+1} - \theta_t) =(1-\beta)\theta_t + \beta(\theta_t - \theta_{t-1})$$
$$- \beta(\alpha_t - \alpha_{t-1})m_{t-1} - \alpha_t(1-\beta)h_t$$

Divide both sides by $1-\beta$
$$\theta_{t+1} + \frac{\beta}{1-\beta}(\theta_{t+1} - \theta_t) =\theta_t + \frac{\beta}{1-\beta}(\theta_t - \theta_{t-1})$$
$$- \frac{\beta}{1-\beta}(\alpha_t - \alpha_{t-1})m_{t-1} - \alpha_t h_t$$

$\square$

**Lemma C.19.** *Suppose that the conditions in Theorem C.4 hold, then*

$$E\left[f(z_{t+1}) - f(z_1)\right] \le \sum_{i=1}^{4} T_i,$$

*where*

$$T_1 = -E\left[\sum_{i=1}^{t}\langle \nabla f(z_i), \frac{\beta_1}{1-\beta_1}(\alpha_i - \alpha_{i-1})m_{i-1}\rangle\right]$$

$$T_2 = -E\left[\sum_{i=1}^{t}\alpha_i\langle \nabla f(z_i), h_i\rangle\right]$$

$$T_3 = E\left[\sum_{i=1}^{t} L\left\|\frac{\beta}{1-\beta}(\alpha_i - \alpha_{i-1})m_{i-1}\right\|^2\right]$$

$$T_4 = E\left[\sum_{i=1}^{t} L\|\alpha_i h_i\|^2\right]$$

*Proof.* By the Lipschitz smoothness of $\nabla f$,

$$f(z_{t+1}) \le f(z_t) + \langle \nabla f(z_t), z_{t+1} - z_t\rangle + \frac{L}{2}\|z_{t+1} - z_t\|^2,$$

Based on (C.18),we have

$$E[f(z_{t+1}) - f(z_1)] =E\left[\sum_{i=1}^{t} f(z_{i+1}) - f(z_i)\right]$$
$$\le E\left[\sum_{i=1}^{t}\langle \nabla f(z_i), z_{i+1} - z_i\rangle + \frac{L}{2}\|z_{i+1} - z_i\|^2\right]$$
$$= -E\left[\sum_{i=1}^{t}\langle \nabla f(z_i), \frac{\beta}{1-\beta}(\alpha_i - \alpha_{i-1})m_{i-1}\rangle\right]$$
$$- E\left[\sum_{i=1}^{t}\alpha_i\langle \nabla f(z_i), h_i\rangle\right] + E\left[\sum_{i=1}^{t}\frac{L}{2}\|z_{i+1} - z_i\|^2\right]$$

Then, using inequality $\|a+b\|^2 \le 2\|a\|^2 + 2\|b\|^2$ and combined with lemma C.18,

$$E\left[\sum_{i=1}^{t}\frac{L}{2}\|z_{i+1} - z_i\|^2\right] \le T_3 + T_4$$

$\square$

**Lemma C.20.** *In this part, we bound $T_1, T_2, T_3, T_4$*

*Proof.* •*Bound for $T_1$*

$$T_1 \le E\left[\sum_{i=1}^{t}\|\nabla f(z_i)\|\|m_{i-1}\|\frac{\beta}{1-\beta}|\alpha_i - \alpha_{i-1}|\right]$$

$$\le H^2\frac{\beta}{1-\beta}E\left[\sum_{i=1}^{t}|\alpha_i - \alpha_{i-1}|\right]$$

$$\le H^2\frac{\beta}{1-\beta}\alpha$$

where the second and last inequality is due to the monotone decreasing property of $\alpha_i$
•*Bound for $T_3$*

$$T_3 \le \left(\frac{\beta}{1-\beta}\right)^2 LH^2 E\left[\sum_{i=1}^{t}(\alpha_i - \alpha_{i-1})^2\right]$$

$$\le 2\alpha\left(\frac{\beta}{1-\beta}\right)^2 LH^2 E\left[\sum_{i=1}^{t}|\alpha_i - \alpha_{i-1}|\right]$$

$$\le 2\alpha^2\left(\frac{\beta}{1-\beta}\right)^2 LH^2$$

where the monotone decreasing property of $\alpha_i$ is also used
•*Bound for $T_4$*

$$T_4 \le H^2 L\alpha^2 E\left[\sum_{i=1}^{t}\frac{1}{t}\right] \le H^2 L\alpha^2(1+logT)$$

where the second inequality is due to $\sum_{i=1}^{t}\frac{1}{t} \le 1+logT$
•*Bound for $T_2$*

$$T_2 = - E\left[\sum_{i=1}^{t}\alpha_i\langle\nabla f(\theta_i), h_i\rangle\right]$$

$$- E\left[\sum_{i=1}^{t}\langle\nabla f(z_i) - \nabla f(\theta_i), h_i\rangle\right] \qquad (42)$$

The second term of (42) can be bounded as

$$- E\left[\sum_{i=1}^{t}\langle\nabla f(z_i) - \nabla f(\theta_i), h_i\rangle\right]$$

$$\le E\left[\sum_{i=1}^{t}\frac{1}{2}\|\nabla f(z_i) - \nabla f(\theta_i)\|^2 + \frac{1}{2}\|\alpha_i h_i\|^2\right]$$

$$\le \frac{L^2}{2}E\left[\sum_{i=1}^{t}\|\frac{\beta}{1-\beta}\alpha_{i-1}m_{i-1}\|^2\right] + \frac{1}{2}E\left[\sum_{i=1}^{t}\|\alpha_i h_i\|^2\right]$$

$$\le \frac{\alpha^2 H^2 L^2}{2}\left(\frac{\beta}{1-\beta}\right)^2\sum_{i=1}^{t}\frac{1}{t} + \frac{\alpha^2 H^2}{2}\sum_{i=1}^{t}\frac{1}{t}$$

$$\le \frac{\alpha^2 H^2}{2}\left[L^2\left(\frac{\beta}{1-\beta}\right)^2 + 1\right](1+logT)$$

where the second inequality is due to $\|\nabla f(z_i) - \nabla f(\theta_i)\| \leq L\|z_i - \theta_i\|$.
Then, consider the first term of (42)

$$
E\left[\sum_{i=1}^{t} \alpha_i \langle \nabla f(\theta_i), h_i \rangle\right]
$$

$$
= E\left[\sum_{i=1}^{t} \alpha_i \langle \nabla f(\theta_i), \kappa g_i + \mu I_i \rangle\right]
$$

$$
\approx \kappa E\left[\sum_{i=1}^{t} \alpha_i \langle \nabla f(\theta_i), \nabla f(\theta_i) + \delta_i \rangle\right] + \mu E\left[\sum_{i=1}^{t} \alpha_i \langle \nabla f(\theta_i), \nabla f(\theta_i) + \delta_i \rangle\right]
$$

$$
= (\kappa + \mu) E\left[\sum_{i=1}^{t} \alpha_i \langle \nabla f(\theta_i), \nabla f(\theta_i) \rangle\right]
$$

The second and third equality holds for the follow reasons: on the one hand, $g_t = \nabla f(\theta_t) + \delta_t$ in which $E[\delta_t] = 0$, so according to (Chen et al., 2018), given $\theta_i$, $E[\delta_i|\theta_i] = 0$; On the other hand, suppose the optimizer runs for a long time, the bias of EMA is small (Zhuang et al., 2020), thus $E(I_t)$ approaches $E(g_t)$ as step increases. Finally, we can finally bound $T_2$

$$
T_2 \leq \frac{\alpha^2 H^2}{2}\left[L^2\left(\frac{\beta}{1-\beta}\right)^2 + 1\right](1 + logT) + (\kappa + \mu)E\left[\sum_{i=1}^{t} \alpha_i \langle \nabla f(\theta_i), \nabla f(\theta_i) \rangle\right]
$$

$\square$

**Theorem C.21.** *(Convergence of* ADMETAS *in non-convex stochastic optimization)*
*Under the assumptions:*

- *$\nabla f$ exits and is Lipschitz-continuous,i.e, $\|\nabla f(x) - \nabla f(y)\| \leq L\|x - y\|$, $\forall x, y$; $f$ is also lower bounded.*

- *At step $t$, the algorithm can access a bounded noisy gradient $g_t$, and the true gradient $\nabla f$ is also bounded.*

- *The noisy gradient is unbiased, and has independent noise, i.e. $g_t = \nabla f(\theta_t) + \delta_t$, $\mathbb{E}[\delta_t] = 0$ and $\delta_i \perp \delta_j, \forall i \neq j$.*

*And $\alpha_t = \alpha/\sqrt{t}$, then for any T we have:*

$$
\min_{t \in [T]} \mathbb{E}\left\|\nabla f(\theta_t)\right\|^2 \leq \frac{1}{\sqrt{T}}(Q_1^{'} + Q_2^{'} logT)
$$

*where $Q_1^{'}$ and $Q_2^{'}$ are constants independent of T.*

*Proof.* We combine lemma C.18, lemma C.19 and lemma C.20 to bound the overall expected descent of the objective. First, we have

$$
E\left[f(z_{t+1}) - f(z_1)\right] \leq T_1 + T_2 + T_3 + T_4 \tag{43}
$$

$$
\leq H^2 \frac{\beta}{1-\beta}\alpha + \frac{\alpha^2 H^2}{2}\left[L^2\left(\frac{\beta}{1-\beta}\right)^2 + 1\right](1 + logT) \tag{44}
$$

$$
- (\kappa + \mu)E\left[\sum_{i=1}^{t} \alpha_i \langle \nabla f(\theta_i), \nabla f(\theta_i) \rangle\right] \tag{45}
$$

$$
+ 2\alpha^2\left(\frac{\beta}{1-\beta}\right)^2 LH^2 + H^2 L\alpha^2(1 + logT) \tag{46}
$$

Notice that

$$E\left[\sum_{t=1}^{T}\alpha_i\langle\nabla f(\theta_t),\nabla f(\theta_t)\rangle\right] \geq E\left[\sum_{t=1}^{T}\frac{1}{\sqrt{t}}\|\nabla f(\theta_t)\|^2\right] \geq \sqrt{T}\min_{t\in[T]}E\left[\|\nabla f(\theta_t)\|^2\right] \quad (47)$$

Rearrange (43), combined with (47) and notice that $\kappa + \mu > 0$, we have

$$\begin{aligned}
\min_{t\in[T]} E\left[\|\nabla f(\theta_t)\|^2\right] \leq &\frac{1}{\sqrt{T}}E\left[\sum_{t=1}^{T}\alpha_i\langle\nabla f(\theta_t),\nabla f(\theta_t)\rangle\right] \\
\leq &\frac{1}{\sqrt{T}}\left[\frac{1}{\kappa+\mu}\left(\frac{\alpha^2 H^2 L^2}{2}\left(\frac{\beta}{1-\beta}\right)^2 + \frac{\alpha^2 H^2}{2} + H^2 L\alpha^2\right)(1+logT)\right. \\
&\left. + \frac{1}{\kappa+\mu}\left(H^2\frac{\beta}{1-\beta}\alpha + 2\alpha^2\left(\frac{\beta}{1-\beta}\right)^2 LH^2 + E[f(z_1)-f(z^*)]\right)\right] \\
= &\frac{1}{\sqrt{T}}(Q_1' + Q_2' logT)
\end{aligned}$$

where $z^*$ is the optimal of $f$, i.e. $z^* = \arg\min_{z} f(z)$

This completes the proof. $\qquad\square$

## C.6 CONVERGENCE ANALYSIS OF FORWARD-LOOKING

In this section, based on Wang et al. (2020), we further analysis forward-looking part to complete the convergence proof of ADMETA optimizer.

According to (Zhang et al., 2019), Lookahead is an algorithm that can be combined with any standard optimization method. The same is true for dynamic lookahead method in forward-looking part. What's more, optimizers with forward-looking is essentially processing with two loops as discussed in the main text. The fast weight is updated by optimizers, while the slow weight is updated by interpolating with fast weight every given period. In other words, the slow weight is updated passively. Therefore, though the slow weight is relevant to optimizers, it is almost irrelevant to the selection of optimizers. For this reason, we only prove the convergence of forward-looking of ADMETAS, which can be easily extended to the ADMETAR.

**Remarks:**(some preliminaries)
Based on the design of the asymptotic dynamic weight $\eta_t$ of the forward-looking part, it can be concluded that when it runs for a long time, $\eta_t$ is highly close to the set point, at which we can safely assume that $\eta_t$ is a constant and thus we denote it as $\eta$. In this way, the analysis of a dynamic lookahead is the same as the case of static lookahead.

According to algorithm of ADMETA, the slow weight $\phi_t$ updates every k steps. We can assume that the slow weight is trained in sync with fast weight. For this purpose, all we should do is to stipulate $\phi_{\tau k+l} = \phi_{\tau k}$, where $k$ denotes the synchronization period, $\tau \in \mathbb{N}^*$ and $0 \leq l < k$.

Define $y_t = \eta\theta_t + (1-\eta)\theta_t$, then according to the update of $\theta_t$ and $\phi_t$, we have

$$y_{t+1} = y_t - \eta\alpha_t m_t$$

and on each period of synchronization, we have

$$\begin{aligned}
y_{\tau k} - \theta_{\tau k} &= (1-\eta)(\phi_{\tau k} - \theta_{\tau k}) = 0 \\
y_{\tau k} - \phi_{\tau k} &= \eta(\theta_{\tau k} - \phi_{\tau k}) = 0
\end{aligned}$$

**Theorem C.22.** *(convergence of forward-looking part)*
*Suppose $f(.)$ is L-smooth, i.e, $\|\nabla f(x) - \nabla f(y)\| \leq L\|x-y\|$, $\forall x,y$. The bias of noisy gradient is bounded, i.e., $|\delta_t| \leq \sigma$, where $\delta_t = \nabla f(\theta_t) - g_t$. Then we have that:*

$$\frac{1}{T}\sum_{t=0}^{T}\mathbb{E}\left\|\nabla f(\theta_t)\right\|^2 \leq \mathcal{O}(\frac{1}{\sqrt{T}})$$

*Proof.* Following the L-smooth property, we have

$$f(y_{t+1}) - f(y_t) \leq -\eta\alpha_t \langle \nabla f(y_t), m_t \rangle + \frac{\eta^2 \alpha_t^2 L}{2} \|m_t\|^2 \tag{48}$$

Taking the expectation of both sides,

$$\mathbb{E}[\langle \nabla f(y_t), m_t \rangle] = \mathbb{E}[\langle \nabla f(y_t), \kappa g_t + \mu I_t \rangle] = \kappa \mathbb{E}[\langle \nabla f(y_t), g_t \rangle] + \mu \mathbb{E}[\langle \nabla f(y_t), I_t \rangle] \tag{49}$$

Consider the term with $\kappa$,

$$\begin{aligned}
\mathbb{E}[\langle \nabla f(y_t), g_t \rangle] &= \langle \nabla f(y_t), \nabla f(\theta_t) \rangle \\
&= \frac{1}{2}[\|\nabla f(y_t)\|^2 + \|\nabla f(\theta_t)\|^2 - \|\nabla f(y_t) - \nabla f(\theta_t)\|^2] \\
&\geq \frac{1}{2}[\|\nabla f(y_t)\|^2 + \|\nabla f(\theta_t)\|^2 - L^2\|y_t - \theta_t\|^2] \\
&= \frac{1}{2}[\|\nabla f(y_t)\|^2 + \|\nabla f(\theta_t)\|^2 - (1-\eta)^2 L^2\|\phi_t - \theta_t\|^2]
\end{aligned} \tag{50}$$

Suppose the optimizer runs for a long time, the bias of EMA is small enough, thus $E(I_t)$ approaches $E(g_t)$. For this reason, we can estimate the term with $\mu$ in (49) the same way as (50).

Based on the bounded bias gradient assumption and inequality that $(a+b)^2 \leq 2a^2 + 2b^2$, we have:

$$\mathbb{E}[\|m_t\|^2] \leq 2\mu^2\mathbb{E}[\|I_t\|^2]\| + 2\kappa^2\mathbb{E}[\|g_t\|^2]\| \leq 4(\mu^2 + \kappa^2)\mathbb{E}\|\nabla f(\theta_t)\|^2 + 4(\mu^2 + \kappa^2)\sigma^2 \tag{51}$$

Combined with (48), (49), (50) and (51), rearrange the inequality and take the expectation

$$\begin{aligned}
\mathbb{E}[f(y_{t+1})] \leq & \mathbb{E}[f(y_t)] - \frac{\eta\alpha_t(\mu + \kappa)}{2}\mathbb{E}[\|\nabla f(y_t)\|^2] - \frac{\eta\alpha_t(\mu + \kappa)}{2}\mathbb{E}[\|\nabla f(\theta_t)\|^2] \\
& + \frac{\eta\alpha_t(1-\eta)^2 L^2(\mu + \kappa)}{2}\mathbb{E}[\|\phi_t - \theta_t\|^2] + 2(\mu^2 + \kappa^2)\eta^2\alpha_t^2 L\mathbb{E}[\|\nabla f(\theta_t)\|^2] \\
& + 2(\mu^2 + \kappa^2)\eta^2\alpha_t^2 L\sigma^2
\end{aligned}$$

Since the learning rate is decreasing to zero, we can safely assume that after several iterations, $1 - \eta\alpha_t L > 0$. Then, summing over one outer loop

$$\begin{aligned}
& \mathbb{E}[f(y_{(\tau+1)k})] - \mathbb{E}[f(y_{\tau k})] \\
\leq & -\frac{\eta\alpha_{(\tau+1)k}(\mu + \kappa)}{2}\sum_{l=0}^{k-1}\mathbb{E}[\|\nabla f(y_{\tau k+l})\|^2] + 2(\mu^2 + \kappa^2)k\eta^2\alpha_{\tau k}^2 L\sigma^2 \\
& - \frac{\eta\alpha_{(\tau+1)k}(\mu + \kappa - 4(\mu^2 + \kappa^2)\eta\alpha_{(\tau+1)k}L)}{2}\sum_{l=0}^{k-1}\mathbb{E}[\|\nabla f(\theta_{\tau k+l})\|^2] \\
& + \frac{\eta\alpha_{\tau k}(1-\eta)^2 L^2(\mu + \kappa)}{2}\sum_{l=0}^{k-1}\mathbb{E}[\|\phi_{\tau k+l} - \theta_{\tau k+l}\|^2]
\end{aligned} \tag{52}$$

Consider the last term of (52), we have

$$\mathbb{E}[\|\phi_{\tau k+l} - \theta_{\tau k+l}\|^2] = \mathbb{E}[\|\theta_{\tau k} - \theta_{\tau k+l}\|^2] \le \alpha_{\tau k}^2 \mathbb{E}\left[\|\sum_{j=0}^{l-1} m_{\tau k+j}\|^2\right]$$

$$=2\kappa^2\alpha_{\tau k}^2 \mathbb{E}\left[\|\sum_{j=0}^{l-1} g_{\tau k+j}\|^2\right] + 2\mu^2\alpha_{\tau k}^2 \mathbb{E}\left[\|\sum_{j=0}^{l-1} I_{\tau k+j}\|^2\right]$$

$$\le 4\kappa^2\alpha_{\tau k}^2 \mathbb{E}\left[\left\|\sum_{j=0}^{l-1}(g_{\tau k+j} - \nabla f(\theta_{\tau k+j}))\right\|^2\right] + 4\kappa^2\alpha_{\tau k}^2 \mathbb{E}\left[\left\|\sum_{j=0}^{l-1}\nabla f(\theta_{\tau k+j})\right\|^2\right]$$

$$+ 4\mu^2\alpha_{\tau k}^2 \mathbb{E}\left[\left\|\sum_{j=0}^{l-1}(I_{\tau k+j} - \nabla f(\theta_{\tau k+j}))\right\|^2\right] + 4\mu^2\alpha_{\tau k}^2 \mathbb{E}\left[\left\|\sum_{j=0}^{l-1}\nabla f(\theta_{\tau k+j})\right\|^2\right]$$

$$\le 4(\kappa^2 + \mu^2)\sigma^2 l\alpha_{\tau k}^2 + 4(\mu^2 + \kappa^2)\alpha_{\tau k}^2 \mathbb{E}\left[\left\|\sum_{j=0}^{l-1}\nabla f(\theta_{\tau k+j})\right\|^2\right]$$

$$\le 4(\kappa^2 + \mu^2)\sigma^2 l\alpha_{\tau k}^2 + 4(\mu^2 + \kappa^2)l\alpha_{\tau k}^2 \sum_{j=0}^{l-1}\mathbb{E}[\|\nabla f(\theta_{\tau k+j})\|^2]$$

where the first equality using the property that $\theta_{\tau k} = \phi_{\tau k} = \phi_{\tau k+l}$.

Summing from $l = 0$ to $l = k - 1$, we get,

$$\sum_{l=0}^{k-1}\mathbb{E}[\|\phi_{\tau k+l} - \theta_{\tau k+l}\|^2]$$

$$\le 2(\kappa^2 + \mu^2)\sigma^2\alpha_{\tau k}^2 k(k-1) + 4(\mu^2 + \kappa^2)\alpha_{\tau k}^2 \sum_{l=0}^{k-1} l \sum_{j=0}^{l-1}\mathbb{E}[\|\nabla f(\theta_{\tau k+j})\|^2]$$

$$= 2(\kappa^2 + \mu^2)\sigma^2\alpha_{\tau k}^2 k(k-1) + 4(\mu^2 + \kappa^2)\alpha_{\tau k}^2 \sum_{j=0}^{k-2}\mathbb{E}[\|\nabla f(\theta_{\tau k+j})\|^2] \sum_{l=j+1}^{k-1} l$$

$$= 2(\kappa^2 + \mu^2)\sigma^2\alpha_{\tau k}^2 k(k-1) + 2(\mu^2 + \kappa^2)\alpha_{\tau k}^2 \sum_{j=0}^{k-2}\mathbb{E}[\|\nabla f(\theta_{\tau k+j})\|^2](j+k)(k-j-1)$$

$(j + k)(k - j - 1)$ achieves its maximal value when $j = 0$. Therefore, we have

$$\sum_{l=0}^{k-1}\mathbb{E}[\|\phi_{\tau k+l} - \theta_{\tau k+l}\|^2]$$

$$\le 2(\kappa^2 + \mu^2)\sigma^2\alpha_{\tau k}^2 k(k-1) + 2(\mu^2 + \kappa^2)\alpha_{\tau k}^2 k(k-1) \sum_{j=0}^{k-2}\mathbb{E}[\|\nabla f(\theta_{\tau k+j})\|^2]$$

Here, we can finally bound the the last term of (52)

$$\mathbb{E}[f(y_{(\tau+1)k})] - \mathbb{E}[f(y_{\tau k})]$$

$$\le -\frac{\eta\alpha_{(\tau+1)k}(\mu+\kappa)}{2} \sum_{l=0}^{k-1}\mathbb{E}[\|\nabla f(y_{\tau k+l})\|^2] + G + M \sum_{l=0}^{k-1}\mathbb{E}[\|\nabla f(\theta_{\tau k+l})\|^2]$$

$$\le -\frac{\eta\alpha_{(\tau+1)k}(\mu+\kappa)}{2} \sum_{l=0}^{k-1}\mathbb{E}[\|\nabla f(y_{\tau k+l})\|^2] + G \tag{53}$$

where

$$G = 2(\mu^2 + \kappa^2)k\eta^2\alpha_{\tau k}^2 L\sigma^2 + (\kappa^2 + \mu^2)(\kappa + \mu)\eta(1 - \eta)^2 L^2\sigma^2 k(k - 1)\alpha_{\tau k}^3$$

and

$$M = -\frac{\eta\alpha_{(\tau+1)k}(\mu + \kappa - 4(\mu^2 + \kappa^2)\eta\alpha_{(\tau+1)k}L)}{2}$$
$$+ (\kappa^2 + \mu^2)(\kappa + \mu)\eta(1 - \eta)^2 L^2\sigma^2 k(k - 1)\alpha_{\tau k}^3$$

When $\alpha$ is small enough, M is below zero, for which the second inequality of (53) holds.

Summing from $\tau = 0$ to $\tau = \Upsilon - 1$, we get

$$\mathbb{E}[f(y_{\Upsilon k})] - \mathbb{E}[f(y_0)]$$
$$\leq -\frac{\eta(\mu + \kappa)}{2}\sum_{\tau=0}^{\Upsilon-1}\alpha_{(\tau+1)k}\sum_{l=0}^{k-1}\mathbb{E}[\|\nabla f(y_{\tau k+l})\|^2] + 2(\mu^2 + \kappa^2)k\eta^2 L\sigma^2\sum_{\tau=0}^{\Upsilon-1}\alpha_{\tau k}^2$$
$$+ (\kappa^2 + \mu^2)(\kappa + \mu)\eta(1 - \eta)^2 L^2\sigma^2 k(k - 1)\sum_{\tau=0}^{\Upsilon-1}\alpha_{\tau k}^3$$

Following Wang et al. (2020), we first assume the learning rate $\alpha$ as a fixed constant, then rearrange the inequality above, we get

$$\frac{1}{\Upsilon k}\sum_{\tau=0}^{\Upsilon-1}\sum_{l=0}^{k-1}\mathbb{E}[\|\nabla f(y_{\tau k+l})\|^2] \leq \frac{2[f(y_0) - f_{inf}]}{\eta\alpha\Upsilon k(\mu + \kappa)} + \frac{4(\mu^2 + \kappa^2)\eta\alpha L\sigma^2}{\mu + \kappa}$$
$$+ 2(\kappa^2 + \mu^2)(1 - \eta)^2\alpha^2 L^2\sigma^2(k - 1)$$

Define T as $\Upsilon k$ and set the learning rate $\alpha$ to $1/\sqrt{T}$

$$\frac{1}{T}\sum_{t=0}^{T-1}\mathbb{E}[\|\nabla f(y_t)\|^2] \leq \frac{2[f(y_0) - f_{inf}]}{\eta\sqrt{T}(\mu + \kappa)} + \frac{4(\mu^2 + \kappa^2)\eta L\sigma^2}{(\mu + \kappa)\sqrt{T}}$$
$$+ \frac{2(\kappa^2 + \mu^2)(1 - \eta)^2 L^2\sigma^2(k - 1)}{T}$$
$$= \mathcal{O}(\frac{1}{\sqrt{T}})$$

$\square$

# D ANALYSIS OF CONVERGENCE RATE

For convex situation, we adopt the regret function to estimate the convergence rate. And for non-convex situation, we adopt the minimum of the expectation of the squared gradient to estimate the convergence, which are corresponding to the proof of convergence since the process of the convergence proof is actually the process of finding the convergence rate.

From Table 6, we notice that the convergence rates for all optimizers for convex case are of magnitude of $\mathcal{O}(1/\sqrt{T})$ and for non-convex are of $\mathcal{O}(logT/\sqrt{T})$, which means in essence, algorithms based on gradient decent follows a similar rate constraint. However, the convergence speed of different optimizers may attribute to many other factors, such as on the implementation. Therefore additional statistical experiments are needed for analysis, as we did in Table 4.

# E EXPERIMENTAL DETAILS

## E.1 HYPERPARAMETER TUNING

For ADMETA optimizer, we first determined a rough value range for learning rate and lambda with the toy model according to the visualization as in Figure B. While for other baseline optimizers, we refer to the recommended/default hyperparameter settings in the original paper. In this way, we get the rouge range of the hyperparameter in optimizers. Then, we search the hyperparameters in the adjacent interval, which is listed in the following three subsections.

Table 6: The comparison of convergence rate of several optimizers.

| Case | Optim | Source | Convergence rate (a rough estimation) |
|---|---|---|---|
| Convex | SGD | Zinkevich (2003) | $\frac{D_\infty^2}{2\alpha T} + \frac{G_\infty^2}{2}\sum_{t=1}^T \alpha_t$ |
| | AMSGrad | Reddi et al. (2019) | $\frac{D_\infty^2 \sqrt{T}}{\alpha(1-\beta_1)}\sum_{i=1}^d \hat{v}_{T,i}^1/2 + \frac{D_\infty}{2(1-\beta_1)}\sum_{t=1}^T\sum_{i=1}^d \frac{\beta \hat{v}_{t,i}^1/2}{\alpha_t}$ $+\frac{\alpha\sqrt{1+logT}}{(1-\beta_1)^2(1-\gamma)\sqrt{1-\beta_2}}\sum_{i=1}^d \|g_{1:T,i}\|$ |
| | ADMETAS | - | $\left(\frac{\beta}{1-\beta}+2\right)\frac{dD_\infty}{4\alpha T} + \left(\frac{2\alpha\beta}{1-\beta}+\alpha\right)dG_\infty^2\sqrt{T}$ |
| | ADMETAR | - | $\frac{(2-\beta_1)D_\infty^2\sqrt{T}}{4\alpha(1-\beta_1)}\sum_{i=1}^d \hat{v}_{T,i}^{1/2} + \frac{(2+\beta_1)\alpha\sqrt{1+\log T}}{2\sqrt{(1-\beta_2)(1-\gamma)}}\sum_{i=1}^d \|h_{1:T,i}\|_2$ |
| Non-convex | SGD | - | - |
| | AMSGrad | Chen et al. (2018) | $\frac{H}{\sqrt{T}}\left(C_1\frac{H^2}{c^2}(1+\log T)+C_2\frac{d}{c}+C_3\frac{d}{c^2}+C_4\right)$ |
| | ADMETAS | - | $\frac{1}{\sqrt{T}}\left[\frac{1}{\kappa+\mu}\left(\frac{\alpha^2H^2L^2}{2}\left(\frac{\beta}{1-\beta}\right)^2+\frac{\alpha^2H^2}{2}+H^2L\alpha^2\right)(1+logT)\right.$ $\left.+\frac{1}{\kappa+\mu}\left(H^2\frac{\beta}{1-\beta}\alpha+2\alpha^2\left(\frac{\beta}{1-\beta}\right)^2LH^2+E[f(z_1)-f(z^*)]\right)\right]$ |
| | ADMETAR | - | $\frac{H}{\sqrt{T}}\left((K_1+K_2)\frac{H^2\alpha^2}{c^2}(1+\log T)+K_3\frac{d\alpha}{c}+K_4\frac{d\alpha^2}{c^2}+K_5\right)$ |

Table 7: Optimizer hyperparameter settings on the CIFAR task.

| Model | task | SGD LR | SGDM LR | Adam LR | RAdam LR | Ranger LR | AdaBelief LR | ADMETAR LR | ADMETAR $\lambda$ | ADMETAS LR | ADMETAS $\beta$ |
|---|---|---|---|---|---|---|---|---|---|---|---|
| ResNet-110 | CIFAR-10 | 0.1 | 0.1 | 0.001 | 0.01 | 0.01 | 0.001 | 0.05 | 0.1 | 0.05 | 0.2 |
| | CIFAR-100 | 0.1 | 0.1 | 0.001 | 0.01 | 0.01 | 0.01 | 0.05 | 0.05 | 0.05 | 0.1 |
| PyramidNet | CIFAR-10 | 0.1 | 0.1 | 0.001 | 0.01 | 0.01 | 0.001 | 0.01 | 0.1 | 0.05 | 0.4 |
| | CIFAR-100 | 0.5 | 0.5 | 0.001 | 0.01 | 0.01 | 0.001 | 0.01 | 0.1 | 0.05 | 0.1 |

### E.2 IMAGE CLASSIFICATION

We conduct image classification experiments on CIFAR-10 and CIFAR-100 datasets, which are trained on a single NVIDIA RTX-3090 GPU. Typical architectures like ResNet-110 and PyramidNet are employed as the baseline models. In the ResNet-110 architecture, there are 54 stacked identical $3 \times 3$ convolutional layers with 54 two-layer Residual Units (He et al., 2016).

While in the PyramidNet architecture, there are 110 layers with a widening factor of 48 (Han et al., 2017). We set the training batch size to 128 and the validation batch size to 256. Both model is trained with 160 epochs. Milestone schedule is adopted as the learning rate decay strategy, with learning rate decaying at the end of 80-th and 120-th epochs by 0.1.

We report the hyperparameters tuning for our proposed ADMETA and other optimizers for reproduction of our experiments. For all optimizers, the weight decay is fixed as $1e - 4$. The searching scheme of hyperparameter settings for each optimizer is concluded as follows:

- For SGD and SGDM, the momentum is fixed as 0.9, and the best-performing learning rate is searched from $\{0.01, 0.05, 0.1\}$ and recommended values in original paper. For our ADMETAS, the $\lambda$ is set to fixed 0.9 and we search the best-performing $\beta$ from $\{0.1, 0.2, 0.3, 0.4\}$ and learning rate from $\{0.01, 0.05, 0.1\}$.

- For all adaptive learning rate optimizers, hyperparameters $\beta_1$, $\beta_2$ and $\epsilon$ are set to $\beta_1 = 0.9$, $\beta_2 = 0.999$ and $\epsilon = $ 1e-9 respectively. For Adam, RAdam and AdaBelief optimizer, the learning rate is searched from $\{0.1, 0.01, 0.001\}$. For Ranger, $\eta$ and $k$ are set to $\eta = 0.5$ and $k = 6$ according to Wright (2019). The learning rate is searched from $\{0.1, 0.01, 0.001\}$. And for our ADMETAR, the setting of $k$ is the same as Ranger, and we search $\lambda$ from $\{0.05, 0.1, 0.2, 0.3, 0.4\}$ and learning rate from $\{0.1, 0.05, 0.01\}$.

The resulting hyperparameters reported in the paper are shown in Table 7, where $LR$ is the abbreviation of learning rate.

Table 8: Optimizer hyperparameter settings on the GLUE benchmark.

| Model | Optim | MNLI | | QQP | | QNLI | | SST-2 | | CoLA | | STS-B | | MRPC | | RTE | |
|---|---|---|---|---|---|---|---|---|---|---|---|---|---|---|---|---|---|
| | | *LR* | $\lambda$ | *LR* | $\lambda$ | *LR* | $\lambda$ | *LR* | $\lambda$ | *LR* | $\lambda$ | *LR* | $\lambda$ | *LR* | $\lambda$ | *LR* | $\lambda$ |
| BERT$_{base}$ | AdamW | 2e-5 | – | 3e-5 | – | 3e-5 | – | 2e-5 | – | 5e-5 | – | 5e-5 | – | 4e-5 | – | 6e-5 | – |
| | RAdam | 2e-5 | – | 2e-5 | – | 6e-5 | – | 4e-5 | – | 1e-4 | – | 4e-4 | – | 1.5e-4 | – | 5e-4 | – |
| | Ranger | 5e-5 | – | 5e-5 | – | 1e-4 | – | 8e-5 | – | 2e-4 | – | 5e-4 | – | 4e-4 | – | 1e-3 | – |
| | AdaBelief | 5e-4 | – | 5e-4 | – | 5e-4 | – | 8e-4 | – | 4e-4 | – | 6e-4 | – | 5e-4 | – | 6e-4 | – |
| | ADMETAR | 1.5e-4 | 0.08 | 1e-4 | 0.36 | 2e-4 | 0.03 | 1e-4 | 0.03 | 7e-4 | 0.02 | 1e-3 | 0.08 | 1.2e-3 | 0.3 | 1.8e-3 | 0.36 |
| BERT$_{large}$ | AdamW | 2e-5 | – | 2e-5 | – | 2e-5 | – | 2e-5 | – | 6e-5 | – | 5e-5 | – | 4e-5 | – | 2e-5 | – |
| | RAdam | 2e-5 | – | 2e-5 | – | 5e-5 | – | 4e-5 | – | 1e-4 | – | 2e-4 | – | 8e-5 | – | 5e-4 | – |
| | Ranger | 5e-5 | – | 5e-5 | – | 5e-5 | – | 6e-5 | – | 6e-5 | – | 5e-4 | – | 5e-4 | – | 5e-4 | – |
| | AdaBelief | 2e-4 | – | 4e-4 | – | 5e-4 | – | 2e-4 | – | 6e-4 | – | 2e-4 | – | 4e-4 | – | 8e-4 | – |
| | ADMETAR | 1.5e-4 | 0.08 | 8e-5 | 0.2 | 8e-5 | 0.03 | 9e-5 | 0.3 | 7e-4 | 0.02 | 1e-3 | 0.03 | 6e-4 | 0.08 | 8e-4 | 0.1 |

Table 9: Hyperparameter settings of SQuAD v1.1 and v2.0 development sets.

| Model | Optim | SQuAD v1.1 | | SQuAD v2.0 | | NER-CoNLL03 | |
|---|---|---|---|---|---|---|---|
| | | *LR* | $\lambda$ | *LR* | $\lambda$ | *LR* | $\lambda$ |
| BERT$_{base}$ | AdamW | 5e-5 | – | 5e-5 | – | 6e-5 | – |
| | RAdam | 1e-4 | – | 5e-5 | – | 5e-5 | – |
| | Ranger | 1e-4 | – | 8e-5 | – | 1e-4 | – |
| | AdaBelief | 1e-3 | – | 8e-4 | – | 5e-4 | – |
| | ADMETAR | 4e-4 | 0.05 | 3e-4 | 0.2 | 2e-4 | 0.3 |
| BERT$_{large}$ | AdamW | 2e-5 | – | 5e-5 | – | 2e-5 | – |
| | RAdam | 6e-5 | – | 5e-5 | – | 3e-5 | – |
| | Ranger | 1e-4 | – | 8e-5 | – | 5e-5 | – |
| | AdaBelief | 8e-4 | – | 8e-4 | – | 4e-4 | – |
| | ADMETAR | 4e-4 | 0.05 | 3e-4 | 0.2 | 1.5e-4 | 0.2 |

### E.3 NATURAL LANGUAGE UNDERSTANDING

In the NLU experiments, we employ a pre-trained language model BERT (Devlin et al., 2018) as our backbone. There are two model sizes for BERT: BERT$_{base}$ and BERT$_{large}$, where the base model size has 12 Transformer layers with 768 hidden size, 12 self-attention heads and 110M model parameters and the large model size has 24 Transformer layers with 1024 hidden size, 16 self-attention heads and 340M parameters.

In natural language understanding, we perform experiments on three modeling types of tasks: text classification, machine reading comprehension and token classification. The text classification uses the GLUE benchmark as the evaluation data set, the machine reading comprehension uses SQuAD v1.1 and v2.0, and the token classification uses the NER-CoNLL03 named entity recognition data set.

We train the eight tasks in GLUE benchmark for 3 epochs on a single NVIDIA RTX-3090 GPU, except for MRPC, which is trained for 5 epochs due to its relatively small data size. The maximum sequence length is set to 128 and the training batch size is set to 32. SQuAD v1.1 and SQuAD v2.0 are trained for 2 epochs with two GPUs. The maximum sequence length is set to 384 and the training batch size per device is set to 12. And NER-CoNLL03 is trained for for 3 epochs on a single GPU. The training batch size per device is set to 8.

Because of the pre-training-fine-tuning paradigm, we only employ the adaptive learning rate optimizer. We set $\beta_1$, $\beta_2$, $\epsilon$ and weight decay of these optimizers to 0.9, 0.999, 1e-8 and 0.0 respectively. $\eta$ and $k$ are set to 0.5 and 6 in the Ranger optimizer and ADMETA uses the same value of $k$ as Ranger. We perform hyperparameter tuning on the learning rate and $\lambda$, and the resulting hyperparameters reported in the paper are shown in Table 8 and 9.

### E.4 AUDIO CLASSIFICATION

Based on Wav2vec (Schneider et al., 2019), the Wav2vec 2.0 (Baevski et al., 2020) is a framework for self-supervised learning of speech representations which is composed of 3 modules: feature

Table 10: Hyperparameter settings of SUPERB and Common Language.

| Optim | SUPERB | | Common Language | |
|---|---|---|---|---|
| | *LR* | $\lambda$ | *LR* | $\lambda$ |
| AdamW | 3e-5 | – | 3e-4 | – |
| AdaBelief | 8e-4 | – | 2e-3 | – |
| Ranger | 3e-4 | – | 5e-4 | – |
| RAdam | 8e-5 | – | 5e-4 | – |
| ADMETAR | 5e-4 | 0.05 | 2e-3 | 0.2 |

encoder, contextualized representations and quantization module. In the feature encoder, there are 7 blocks with temporal convolutions that have 512 channels for each block and a relative positional embeddings of the convolutional layer modeling has kernel size of 128 and 16 groups.

Among the configurations of Wav2vec 2.0, we choose Wav2vec 2.0$_{base}$ model, which has 12 Transformer blocks, 95M parameters and 8 attention heads, with model dimension of 768 and inner dimension (FFN) of 3072. We finetune Wav2vec 2.0$_{base}$ for keyword spotting and language identification on SUPERB dataset (Yang et al., 2021) and Common Language (Sinisetty et al., 2021) dataset respectively. The dataset size of keyword spotting is smaller than that of language identification, so we use a single NVIDIA RTX-3090 GPU for training on the SUPERB dataset, and use four GPUs for parallel training on the Common Language dataset. The keyword spotting model is trained for 5 epochs with training batch size 32 and language identification model for 10 epochs with training batch size 8 per device.

Due to the same reason as in NLU experiments, i.e. the pre-training-fine-tuning paradigm, we only employ adaptive learning rate optimizers here. For all optimizers chosen, we fix $\beta_1 = 0.9, \beta_2 = 0.999, \epsilon = 1e - 8$ and set weight decay to 0.0. The learning rate is searched from {5e-5, 8e-5, 1e-4, 3e-4, 5e-4, 8e-4}, and for ADMETAR, $\lambda$ is searched from {0.05, 0.1 0.2}. The resulting hyperparameters reported in the paper are shown in Table 10.

## F    FUTURE WORK

In the future work, for backward-looking part, though DEMA provides a more flexible way to deal with past gradients, it is still unable to intelligently judge the value of certain historical gradient information, such as discarding some obviously unreasonable gradients caused by noise. A better optimizer may have the ability to forget these wrong information and take advantage of what works, just working like human brains. For forward-looking part, our method takes the constant coefficient into a dynamic one. It is kind of like milestone scheme of learning rate decay strategies to some extent. However, several experiments (Huang et al., 2017; Ma, 2020) have shown that cosine strategy (Loshchilov & Hutter, 2016) works better. Therefore, we will follow the cosine scheme and propose a new forward-looking strategy that may work even better.

## G    PERFORMANCE OF SGDM AND ADMETAS ON FINETUNE SETTING

In this section, we test the performance of SGDM and ADMETAS on fintune setting and the results are shown in Table 11. For keyword spotting (SUPERB) (Yang et al., 2021) task, we train the models for 5 epochs and use Wav2vec$_{base}$ (Schneider et al., 2019) as the baseline model. And for CIFAR-10 (Krizhevsky et al., 2009) task, we train the model for 40 epochs from the checkpoint already trained with Adam using learning rate of 0.001 for 160 epochs. The baseline model of CIFAR-10 is ResNet-110 (He et al., 2016) with deep CNN architecture. We report the results of best hyperparameter settings for SGD and ADMETAS via grid searching.

From Table 11, we notice that in SUPERB task, compared to adaptive learning rate methods, SGDM achieves worse results in SUPERB task, but not by much, which shows that SGDM can also be used in finetune setting. While ADMETAS can achieve better result than any other learning rate methods used in our experiment, demonstrating the advantage of our approach. This phenomenon contradicts the mainstream view that SGD family is not suitable for finetune task. While for CIFAR-10 task, SGDM and ADMETAS both improve the performance compared to the start point. However, they

Table 11: Performance of SGDM and ADMETAS on finetune setting.

| Optimizer | SUPERB | CIFAR-10 |
|---|---|---|
| SGDM | 98.25 | 91.71 |
| **ADMETAS** | 98.54 | 91.87 |

Table 12: Performance of SGD family optimizers in CIFAR task.

| Optim | CIFAR-10 | CIFAR-100 |
|---|---|---|
| **ADMETAS** | 94.12 | 73.74 |
| SGDM | 93.68 | 72.07 |
| SGDM (lr=0.5) | 93.65 | 73.48 |

are both obviously worse than the performance of training the task from scratch using SGDM and ADMETAS respectively, which shows that pre-training is a very strong approach that makes the model achieve a good state.

The reason why ADMETAS performs better than SGDM in finetune setting may lie in two aspects. On the one hand, DEMA scheme in the backward-looking part reduces the overshoot problem that may do harm especially near convergence. On the other hand, the forward-looking part improves the stability of the training process.

# H    INFLUENCE OF DIFFERENT LEARNING RATES IN SGD FAMILY OPTIMIZERS

Since the learning rate of 0.5 for SGDM is a recommended value in Han et al. (2017) but not in (He et al., 2016), to alleviate the influence of different learning rates, we also try the performance of SGDM with a learning rate of 0.5 in the ResNet-110 network and the results are listed in Table 12.

The results show that choosing a large learning rate for SGDM may increase the performance, as shown that when setting the learning rate to 0.5 instead of 0.1, the recommended value in ResNet-110. However, this is not always true since the performance on CIFAR-10 when using the learning rate of 0.5 does not get prompted.

