# OpenReview forum: "Admeta: A Novel Double Exponential Moving Average to Adaptive and Non-adaptive Momentum Optimizers with Bidirectional Looking"
_ICLR.cc/2023/Conference — Submitted to ICLR 2023_

### Official Review · Reviewer_kSFP · 2022-10-21

**Confidence:** 3
**Correctness:** 2
**Technical Novelty And Significance:** 2
**Empirical Novelty And Significance:** 1
**Recommendation:** 6

**Clarity, Quality, Novelty And Reproducibility:**

The article is clearly written and presents an enlightening method. I think the attached code can be reproduced.

**Strength And Weaknesses:**

Strength:
1. The paper provides a point of view that is often overlooked by researchers.
2. This paper provides experimental results on various tasks and datasets to demonstrate the advantages of the proposed method.
3. The paper is easy to read.

Weaknesses:

1. The proof in the paper, although valid, is not sufficient to demonstrate the advantages of the proposed method, as it only guarantees convergence, which is the most basic requirement of an optimizer in deep learning.

2. The paper can be improved by adding more theoretical explanations. What's more, the paper could add more and larger datasets.

3. The improvements of the proposed method on some tasks are small in the experiments, so error bars are needed to verify that these improvements are not due to randomness. However, this paper does not provide error bars for the empirical results.

4. AdaBelief's report in CIFAR seems bad. However, in the original paper by Adabelief ([1]), it achieves state-of-the-art results, albeit with a different backbone model. I don't know if there is a problem with the parameter settings of AdaBelief.

5. SGDM in PyramidNet tried a learning rate of 0.5. Why not try it in ResNet110?

**Summary Of The Paper:**

This article shows that exponential moving averages have several weaknesses, namely momentum and Lookahead optimizers. The authors then propose modifications corresponding to these weaknesses and propose a framework, Admeta, to analyze optimizers. Theoretical results are provided to show the convergence and the reasons for such modifications. Empirical results on various tasks are provided to demonstrate that the proposed method outperforms the baseline optimizer and the currently proposed optimizer.

In the implementation, the authors propose new variants of Adam and SGD using the DEMA method and a dynamic look-ahead strategy to avoid the potential negative effects of traditionally used EMA and look-ahead optimizers. The paper also provides evidence that both implementations converge in the convex and non-convex cases.

**Summary Of The Review:**

The paper is well written and easy to read. To clearly explain these methods, the authors combine text, formulas, and figures. For this, I thank the author for his hard work. I also believe that the Admeta framework provides an impressive perspective on the future work of the optimizer. Experiments demonstrate its good results in many tasks. All in all, it's an interesting piece of work that I'm sure would be of interest to researchers in related fields.

[1] AdaBelief optimizer: adjust step size based on observed gradients

---

> ### Author Response · Authors · 2022-11-09
> **Response to Reviewer kSFP**
>
> We thank the reviewer for the careful reviews and positive comments. Responses to specific questions and concerns are listed below:
>
> W#1) lack of sufficient analysis to demonstrate the advantages of the proposed methods.
>
> Thanks for your review. As mentioned in Section 2.2, we discuss that past gradients in EMA follow a fixed proportionality, which is incompatible with the actual optimization. On this basis, we propose DEMA, which deals with the historical gradients and the current gradient in a more flexible way by further controlling the proportion of past gradients. And in Section 2.3, we analyze the importance of dynamic lookahead for training on different stages (during and near convergence). In addition, in ablation study, we demonstrate the role of each part empirically and we also demonstrate the effectiveness of DEMA visually in Figure 3.
>
>
> W#2) more theoretical explanations are needed. More and larger datasets are needed.
>
> Thank you for the valuable suggestions. During rebuttal period, according to your valuable comments and those of other reviewers, we plan to add more analysis on convergence speed to show our optimization efficiency and update the paper soon.
>
> And for large dataset training settings, evaluating the optimizer on large datasets allows for better verification of the optimization advantage. However, on the one hand, our optimizer is not only designed for large-scale dataset optimization, because the choice of optimizer in large dataset optimization is not so important as there is a lot of labeled information. On the other hand, we have trained a variety of dataset scales on the two modes of training-from-scratch and fine-tuning. This fully demonstrates the effectiveness of our optimizer design. Due to the resource and time constraints, we may not be able to complete the experiments in ImageNet and BERT pre-training stage during the rebuttal period. We will explore this in our future work when more resources are available.
>
> W#3) error bars are needed.
>
> Thank you for the suggestion. According to your suggestion, we will add the error bars to Table 1 in Image Classification. And on other tasks, we followed the practice of previous papers and reported the average results of 3 runs with different seeds.
>
> Q#1) why AdaBelief's report in CIFAR seems bad, since the results are good in the original paper though with a different model
>
> Thanks for your question. We have carried out several groups of experiments on hyperparameters, and ensure that it is not caused by the influence of bad hyperparameters. Our choice of hyperparameters followed the following rule: first, we searched for the recommended/default settings in the original papers of these optimizers: the learning rate for AdaBelief is 0.001, and for RAdam and Ranger are 0.01. For both, \beta_1 and \beta_2 were fixed to 0.9 and 0.999. Then we fixed some hyperparameters, like \beta1 and \beta2 and estimated a searching range which has already been listed in the appendix. Finally, we chose the optimum settings, which have also been listed in the appendix. Therefore, we can only attribute this situation to the fact that AdaBelief does not adapt to the model structure we adopted, or it may be that AdaBelief is not stable enough.
>
> Q#2) try the learning rate of 0.5 of SGDM in ResNet110?
>
> Thank you for your suggestion. We are doing the experiment of trying the learning rate of 0.5 for SGDM in ResNet110. And the results are expected to be available in the update version.

---

### Official Review · Reviewer_merB · 2022-10-23

**Confidence:** 5
**Correctness:** 3
**Technical Novelty And Significance:** 2
**Empirical Novelty And Significance:** 3
**Recommendation:** 6

**Clarity, Quality, Novelty And Reproducibility:**

Clarity:
- The introduction is straightforward and the method is stated clearly.

Quality:
-This work is OK.

Novelty:
- The innovation is a bit incremental.

Reproducibility:
-Checked

**Details Of Ethics Concerns:**

None.

**Strength And Weaknesses:**

Pros:
- The authors identify the problem of the traditional used EMA and Lookahead method and give improvements to modify them.
- The introduction of the method is straightforward and easy to follow and the motivation is very interesting which comes from stock markets.
- It is simple but effective. I am positively surprised by the results achieved in some tasks brought by the modifications.

Concerns:
- I notice that in the CIFAR-100 task trained on PyramidNet, the results of R-Adam, Ranger, and Adabelief are even worse than AdamW. It is very interesting. What hyper-parameters do you use? More discussion of this phenomenon is welcomed.
- Can you explain why SGD-family achieves awful results in Finetune task? Additional experiments are needed to explore this point. How will AdmetaS perform in Finetune tasks? Will all SGD family optimizers behave like this? What if the model chooses the Deep RNN/CNN architecture instead of the Deep Transformer architecture?
- The authors simply changed EMA to a DEMA variant and turn lookahead into a dynamic one. I think innovation is a bit incremental.
- There are some problems in the organization of the paper. Many important parts have been placed in the appendix, which increases the difficulty for readers.


**Summary Of The Paper:**

The paper proposes a framework for stochastic optimizers named Admeta. Based on SGD and R-Adam, the authors provide two implementations, AdmetaS and AdmetaR.

Specifically, the author contributes a new gradient decent algorithm framework in the style of SGD and Adam optimizer. First, the author identifies the problem of the commonly used EMA method: the increasing lag time and overshoot problem. Then the author identifies the problem of the Lookahead optimizer: a slow stepsize around the early stage of training. The paper is well written.


**Summary Of The Review:**

I believe that the paper has a lot of potentials but requires some additional discussion. My main concern lies in the results and innovation of the proposed optimizer. In general, this paper is worth publishing if concerns can be well addressed in the rebuttal stage.

---

> ### Author Response · Authors · 2022-11-09
> **Response to Reviewer merB**
>
> We sincerely thank the reviewer for the valuable comments and suggestions. Responses to specific questions and concerns are listed below:
>
> Q#1）why in the CIFAR-100 task trained on PyramidNet, the results of R-Adam, Ranger, and AdaBelief are even worse than AdamW.
>
> Thanks for your careful review. This phenomenon does exist. We have carried out several groups of experiments on hyperparameters, and it can be seen that it is not caused by the influence of hyperparameters. Our choice of hyperparameters followed the following rule: First, we searched for the recommended/default settings in the original papers of these optimizers: the learning rate for AdaBelief is 0.001, and for RAdam and Ranger are 0.01. For both, \beta_1 and \beta_2 were fixed to 0.9 and 0.999. Then we fixed some hyperparameters, like \beta1 and \beta2 and estimated a searching range which has already been listed in the appendix. Finally, we chose the optimum settings, which have also been listed in the appendix.
>
> As for the reason, compared to ResNet110, PyramidNet has a more complicated structure, so the benefits of optimizers may not be reflected. Therefore, it is possible that AdaBelief, Ranger and RAdam achieve worse results than AdamW. In addition, the improvement of our Admeta optimizer on complicated models also shows the strength of our optimizer.
>
> Q#2）Why SGD-family achieves awful results in Finetune task? How will AdmetaS perform in Finetune tasks? Will all SGD family optimizers behave like this? What if the model chooses the Deep RNN/CNN architecture instead of the Deep Transformer architecture?
>
> Thank you for the question and insightful comments. In our paper, the finetune tasks all employ Transformer models. It is not always true that SGD achieves awful results in Transformer models, and it is not the major focus of this work. According to your suggestion, we will experiment the finetuning performance of SGD-family (including AdmetaS) on Audio Classification task with Transformer model and Image Classification task with Deep CNN model and will soon put it in the appendix. In addition, we have experimented with Deep CNN architecture in Image Classification task and the results are shown in Table 1. In our new experiments, the Transformer can also obtain good results under the SGD family optimizers of SGDM and AdmitaS
>
>
> W#1）the innovativeness of changing EMA to a DEMA variant and turning lookahead into a dynamic one is incremental.
>
> Thanks for your comment. The proposed Admeta optimizer provides a bidirectional-looking view that is often overlooked by other optimizers. Our major contribution is to propose a novel framework, that incorporates forward-looking and backward-looking into model optimization.
> In the backward-looking part, we employ a novel DEMA strategy, which is quite different from original DEMA in [1]. Specifically, we designed innovative double exponential moving average forms with customized coefficients \mu and \kappa. Therefore, our method is not just a variant of EMA, but a novel design. According to the ablation study, the proposed DEMA is important for the improvement of backward-looking part.
> In the forward-looking part, we considered the geometry/curvature of the optimization process as shown in Figure 1, and turned the static weight of lookahead into an asymptotic dynamic one. According to the ablation study we can conclude that the original static weight in lookahead even achieves worse results than baseline RAdam, which highlights the importance of our design.
>
> [1] Smoothing Data With Faster Moving Averages
>
> W#2）there are some problems in the organization of the paper which increases the difficulty for readers.
>
> Thanks for valuable comment. Due to the space constraints, we put some parts in the Appendix. We will reorganize it if more space is available.

---

### Official Review · Reviewer_dVGC · 2022-10-23

**Confidence:** 4
**Correctness:** 3
**Technical Novelty And Significance:** 3
**Empirical Novelty And Significance:** 3
**Recommendation:** 6

**Clarity, Quality, Novelty And Reproducibility:**

Clarity: The paper is well organized and clear.

Quality: See contents above.

Novelty: This article is innovative as a conference paper.

Reproducibility: The paper has released the code for reproducing.


**Strength And Weaknesses:**

Strengths:

The ideas are reasonable, and this work gives a novel view in the optimizer field.

The authors conduct several experiments in Image Classification, Natural Language Understanding and Audio Classification, and experiments show the effectiveness of the proposed method. For all but a few tasks, this method shows advantages over other recent optimizers.

This paper follows a good logic and is easy to understand.

Concerns:

On the other hand, I also have some concerns and questions toward this paper:

 (1) The theoretical explanation for the advantage of DEMA is insufficient. For example, the authors mention that the past gradients in EMA follow a fixed proportionality which is incompatible with actual optimization. Why?

 (2) In Image Classification, why the improvement for ResNet-110 is larger than PyramidNet?

 (3) The authors introduce a bias term in the DEMA method, but omit it in the rest of the paper and point out that this does not make difference. Why?

(4) The choice of $\mu$ and $\kappa$ seems strange. What is the reason for constructing such a function?

(5) How do you choose the hyper-parameter of the Admeta optimizer? It seems that the hyper-parameters chosen for each task are different.

 (6) There are some deficiencies in the theoretical analysis, although I understand the space is limited. If more theoretical analysis is added, it will be more convincing and interesting.

 (7) Other works on optimizers like [1] tend to give an interval on experiment results in the appendix. I suggest that the results need to be tested for statistical significance.

(8) Although training on large datasets is not necessary for the optimizer, if Admeta can be used in the large-scale dataset ImageNet or BERT pre training stage, the advantages of the proposed optimizer will be better demonstrated.

(9) Though the related work is organized well and summarized into SGD Family, Adam Family, Stochastic Second Order Family and Other Optimizers, the authors should clearly explain the relation between the proposed method and related works.

[1] Apollo: An Adaptive Parameter-wise Diagonal Quasi-Newton Method for Nonconvex Stochastic Optimization


**Summary Of The Paper:**

The authors give a clear introduction to current optimizers in deep neural networks and propose a new framework for improvement, which divides the optimizer into two parts, forward-looking and backward-looking.

To replace the traditional EMA method used in backward-looking part, a variant of double exponential moving average (DEMA) is used. The authors demonstrate its advantage over EMA by analyzing the structure of the formula and attaching a comparison figure, which visually presents the convergence trend in the appendix.

And to further improve the Lookahead optimizer, an asymptotic weight is used, which achieves faster training than constant lookahead around the early stage and maintains the advantage of better convergence around the final stage. The authors also provide two alternatives to compute the asymptotic weight.


**Summary Of The Review:**

The paper proposes a novel framework for optimizer. It seems that this work is quite original and the experimental results are relatively good. The main weaknesses are that some points are not made clear and extended experiments are needed to demonstrate the effectiveness of the proposed method. In general, the paper is well-written and comprehensible, and the idea is impressive.

---

> ### Author Response · Authors · 2022-11-09
> **Response to Reviewer dVGC (Part 1)**
>
> We thank the reviewer for the positive reviews and insightful questions. Responses to specific questions and concerns are listed below:
>
> W#1) more words to explain the claim that past gradients in EMA follow a fixed proportionality is incompatible with actual optimization.
>
> Thanks for your insightful comment. Due to the use of minibatch strategy, the input is randomly sampled. The effect of each minibatch towards optimization is varied. Therefore, applying a fixed proportional to past gradients is not a reasonable approach since it does not take into account the changeable process. The disadvantage of overshoot that EMA usually has may also be caused by the above reasons. In this way, we considered using a more flexible ratio, which is account for the design of some coefficients in DEMA. Experiments and visualizations show that our method works well, which reflects that our hypothesis is well-supported.
>
> Q#1) why the improvement for ResNet-110 is larger than PyramidNet?
>
> Thanks for your question. Compared to ResNet110, PyramidNet has a more complicated structure and can achieve better results in these tasks. In cases where the model is strong enough, the selection of optimizer will not be the main factor for the final performance. As shown in Table 1, compared to Adam, RAdam and AdaBelief achieve just a bit of improvement on CIFAR-10 task and even achieve worse results on CIFAR-100 task, which also supports our above claims.
>
> Q#2) why the bias term is omitted in the rest of the paper and the author claims that it makes no difference
>
> Thanks for your careful review. The omission of bias term in the analysis part of paper is because that the bias term is a small amount and goes exponentially to 0. Then, using triangular inequality of norm, we can conclude that it does not affect the estimation of the order. In the implementation, we found that adding this bias term can result in more stable numerical calculation and slightly improve the results, so we applied it in practice.
>
> Q#3) how to construct \mu and \kappa?
>
> Thanks for your question. The coefficients \mu and \kappa are designed by our experience and verified on the toy model by visualization. In fact, the original design of \mu and \kappa is
> $$\mu = 10 (1-\lambda) - 10 \frac{1-\lambda}{\lambda} + 5$$
> $$Κ = 10 \frac{1-\lambda}{\lambda} +1$$
> After simplification, we get the form in the paper.
>
>
> Q#4) how do you choose the hyper-parameter of the Admeta optimizer? It seems hyper-parameters chosen for each task are different.
>
>
> Thanks for your question. As in Section D in Appendix, we briefly introduced the hyperparameter setting and tuning process. We will further describe this hyperparameter tuning process in detail. For our Admeta optimizer, we first determined a rough value range for learning rate and lambda with the toy model according to the visualization as in Figure 3. While for other baseline optimizers, we referred to the recommended/default hyperparameter settings in the original paper. In this way, we got the rouge range of the hyperparameter in optimizers. Then, we searched the hyperparameters in the adjacent interval, which has been listed in the experimental details in the Appendix. We will update the hyperparameter setting part and give a more comprehensive description of the process of parameter tuning soon.
>
> And for the hyperparameters of each task, they were determined according to the related works on these tasks since they depend on the scale of the backbone models and tasks. And we obtained the final hyperparameter by tuning accordingly.
>
>
> W#2) there are some deficiencies in the theoretical analysis.
>
> Thank you for your valuable suggestion. We present the analysis of the fixed proportionality of EMA in Section 2.2, the performance of static and dynamic lookahead based on a toy loss in section 2.3, and a visual update path comparison between EMA and DEMA in Section B in Appendix. According to your suggestion and other reviewers, we will further add the analysis of the convergence speed in addition to the existing theoretical analysis. We hope this can satisfy your request.
>
> W#3) an interval on experiment results is needed
>
> Thank you for this suggestion. To make a fair comparison with previous works in Natural Language Understanding and Audio Classification tasks, we reported the results by taking the average of three different runs. And we will soon add the interval on results for Image Classification tasks. From the current results, compared to baseline, the improvement brought by our method is significant and the improvements are not due to the model variants.

---

> > ### Author Response · Authors · 2022-11-09
> > **Response to Reviewer dVGC (Part 2)**
> >
> > W#4) more experiment on large-scale dataset ImageNet or BERT pre training stage
> >
> > Thank you for your suggestion. Yes, evaluating the optimizer on large datasets allows for better verification of the optimization advantage. However, on the one hand, our optimizer is not only designed for large-scale dataset optimization, because the choice of optimizer in large dataset optimization is not so important as there is a lot of labeled information. On the other hand, we have trained a variety of dataset scales on the two modes of training-from-scratch and fine-tuning. This fully demonstrates the effectiveness of our optimizer design. Due to the resource and time constraints, we may not be able to complete the experiments in ImageNet and BERT pre-training stage during the rebuttal period. We will explore this in our future work when more resources are available.
> >
> >
> > W#5) the authors should clearly explain the relation between the proposed method and related works in the section of related work
> >
> > Thank you for your suggestion. In our framework Admeta, we provide two versions, AdmetaS and AdmetaR, based on SGD and RAdam respectively. In this way, Admeta is related to SGD family and Adam family. The second order family is also introduced because the framework of Admeta can also be applied in this family, but we do not try it on due to the limited resources. We also introduce Lookahead optimizer, since it is the base of our forward-looking part. We will modify the related work part in the revised version and clarify the relationship between Admeta and other optimizers clearly.

---

### Official Review · Reviewer_RS9p · 2022-10-25

**Confidence:** 4
**Correctness:** 4
**Technical Novelty And Significance:** 3
**Empirical Novelty And Significance:** 3
**Recommendation:** 6

**Clarity, Quality, Novelty And Reproducibility:**

The paper is clear for readers to follow with a nice quality and interesting novelty. The reproducibility is good.

**Strength And Weaknesses:**

The strengths of the paper include an interesting framework to analyze the optimizer with modifications to each part, and a nice visalization figure show the comparison between DEMA and EMA. In my views, the limitations of the paper are that some results are not so good, especially in some relatively large models or datasets, such as MNLI with large BERT model. (But it should not be a big problem for the optimizer.)
My detailed comments are listed as below.

1) Theorems 1-4 provide convergence proof on AdmetaR and AdmetaS, the two implementation versions of Admeta based on RAdam and SGD respectively. Before the main proof in the appendix, the authors make many assumptions and omit many things and argue that this argument can be attended to these cases. For example, the authors omit the forward-looking part in the convergence proof.  Are these assumptions and neglections reasonable? More discussion should be put in this part.

2) As in some related works, the author should do some convergence rate analysis of AdmetaR in the convex and nonconvex settings to demonstrate its high convergence speed?

3) The authors should introduce the hyperparameters tuning with more details since it is very important for the application of this optimizer in other tasks.

4) In the ablation study part, I notice that AdmetaS – DEMA achieves worse result than SGDM, which may indicate that DEMA is not always a good method compared to EMA unless combined with a dynamic method. What may be the reason for this? The author needs to explain more.

5) Separating optimizers into forward-looking and backward-looking parts is not a novel view. In fact, it is already used in Ranger ([1]). What is the difference of Admeta compared to Ranger optimizer.


6) I notice that authors use SGD with Nesterov momentum in the experiment. It would be more interesting to see Nadam, Adam with Nesterov momentum in the comparisons. How about incorporating Nesterov’s approach in DEMA?

[1] Less Wright. Ranger - a synergistic optimizer. https://github.com/lessw2020/ Ranger-Deep-Learning-Optimizer, 2019.

**Summary Of The Paper:**

In this paper, the authors found that using a variant of DEMA to replace the EMA used in past optimizers and using a dynamic lookahead optimizer can achieve a better result. Motivated by that, the author proposed a novel optimizer framework Admeta, and implement it on SGD and Radam. Empirical results demonstrate the superior performance of Admeta compared to other optimizers.

**Summary Of The Review:**

This work has introduced a new optimizer with modifications on two parts. Despite some deficiencies in theory, the paper is well written.

---

> ### Author Response · Authors · 2022-11-09
> **Response to Reviewer RS9p (Part 1)**
>
> We thank the reviewer for the positive and detailed review as well as the suggestions for improvement.
>
> W#1) some results not so good, especially in some relatively large models or datasets, such as MNLI with large BERT model.
>
> Thanks for your comment. For MNLI, it is a finetune task with the training dataset of 393k examples. Compared to other GLUE tasks, especially smaller ones, the supervision signals of MNLI are relative more abundant. Thus, when using large backbone models, like large BERT model, the advantage of optimizers is not obvious as in smaller datasets or models. As evidence of our claim, there is a similar phenomenon in experiment that Ranger’s performance is also quite similar and AdaBelief even achieves worse results. While on other tasks with less sufficient supervision signals, the margin between AdmetaR and base optimizers is significant, which demonstrates the advantage of our proposed Admeta optimizers on GLUE tasks.
>
>
> W#2) the authors omit the forward-looking part in the convergence proof, which needs more explanation.
>
> Thank you for valuable suggestion. Due to the space constraints, we did not include the forward-looking part in the convergence proof. In fact, relative theoretical analysis has been discussed in [1], so we omit it for brevity. However, according to your suggestion, we will add more explanation about the convergence of forward-looking part, and add it to section C in Appendix.
>
> [1] LOOKAHEAD CONVERGES TO STATIONARY POINTS OF SMOOTH NON-CONVEX FUNCTIONS
>
> W#3) convergence rate analysis of AdmetaR in the convex and nonconvex settings is needed.
>
> Thanks for the insightful comments. Since for optimizers in deep neural network, the theoretical convergence rate is approximated by the algorithm complexity. It can only provide a rough upper bound instead of accurately reflecting the speed of optimizers. And the actual convergence speed can be affected by multiple factors. On the one hand, lots of inequality scaling arguments are used in relative analysis. The magnitude of scaling and the initial assumptions will probably affect the final theoretical results (The assumptions are not usually the same because of the design of SGD-family and Adam-family optimizers). On the other hand, the convergence rate is affected by implementations etc. Thus, in current version, to compare the actual speed of convergence with other optimizers, we recorded the training time of different optimizers on the Audio Classification task, as shown in Table 4. However, according to your suggestion, we will soon provide a more detailed comparison of convergence rate of optimizers in the Appendix.
>
> W#4) more details about hyperparameter tuning.
>
> Thanks for your comment. As in Section D in Appendix, we briefly introduced the hyperparameter setting and tuning process. We will further describe this hyperparameter tuning process in detail. For our Admeta optimizer, we first determined a rough value range for learning rate and lambda with the toy model according to the visualization as in Figure 3. While for other baseline optimizers, we referred to the recommended/default hyperparameter settings in the original paper. In this way, we got the rouge range of the hyperparameter in optimizers. Then, we searched the hyperparameters in the adjacent interval, which has been listed in the experimental details in the Appendix. We will update the hyperparameter setting part and give a more comprehensive description of the process of parameter tuning soon.
>
>
> Q#1) how to explain the fact that in the ablation study part, AdmetaS – DEMA achieves worse result than SGDM, since it may indicate that DEMA is not always a good method compared to EMA unless combined with a dynamic method.
>
> Thanks for your question. In ablation study, the phenomenon that AdmetaS-DEMA (equals to SGDM+Dynamic Forward-looking) achieves worse result than SGDM actually shows that EMA is not suitable when combined with a dynamic forward-looking part, not that DEMA must be combined with a dynamic method due to DEMA is not used in both AdmetaS-DEMA and SGDM
> AdmetaS-DEMA is worse than SGDM, demonstrating that EMA is not always suitable for forward-looking. While with DEMA, adding forward-looking can further improve the results, which shows that DEMA is effective.
>
>
> To examine the effectiveness of DEMA, we can refer to AdmetaS-LF or AdmetaR-LF and compare them with SGDM and RAdam respectively. We also realize that DEMA may not a method that suits all optimizer family when it is used separately since they achieve similar results with baseline optimizers. For this phenomenon, this may be due to the lack of reference basis for DEMA in the absence of forward exploration, which thus does not bring better optimization results.

---

> > ### Author Response · Authors · 2022-11-09
> > **Response to Reviewer RS9p (Part 2)**
> >
> > Q#2) Ranger also has the view of separating optimizers into forward-looking and backward-looking. What is the difference?
> >
> > Thanks for the question. Our work details the role of forward-looking and backward-looking and takes into account the intrinsic connections of these two components. In backward-looking part, we propose a DEMA scheme to replace the traditional EMA scheme; in forward-looking part, we propose a dynamic lookahead strategy to replace the original static lookahead strategy. However, Ranger, as a baseline of our optimizer, just simply combines RAdam and Lookahead together without considering how to integrate the two components effectively. It simply leverages these components and lacks detailed interaction of their components in training.
> >
> >
> > Q#3) how about incorporating Nesterov’s approach in DEMA?
> >
> > Thanks for this interesting question. The idea of incorporating Nesterov’s approach in DEMA is possibly feasible. However, Nesterov momentum actually modifies the momentum by computing gradient based on the approximation of the next position and thus changing the descent direction, while the DEMA can also be regarded as a method to change the decent direction of vanilla momentum. In this way, they are not orthogonal methods and are thus in conflict so far. During the rebuttal period, we may not have enough time to dig deep into how to integrate them together in an efficient way and we will regard this as a direction for future work.

---

### Author Response · Authors · 2022-11-09
**General Response**

Dear Reviewers and AC,

We sincerely appreciate all the reviews. They give positive and high-quality comments on our paper with a lot of constructive feedback. We are working on incorporating the insightful and valuable suggestions from the reviewers. We will update the draft accordingly soon.

Authors

---

### Author Response · Authors · 2022-11-16
**General Response After Paper Update**

Dear Reviewers and AC,

We have updated our draft to incorporate the insightful suggestions of the reviewers:

- Following Reviewer RS9p’s suggestion, we have added the convergence proof of the forward-looking part in Section C.6.

- Following Reviewer RS9p and dVGC’s suggestion, we have added more details about hyperparameter tuning in Section E.1 and the convergence rate analysis in Section D.

- Following Reviewer dVGC’s suggestion, we have added more details about the analysis of the fixed proportionality of EMA in Section 2.2.

- Following Reviewer dVGC’s suggestion, we have added more clarification to explain the relation between our method and related works in Section A.

- Following Reviewer dVGC and kSFP’s suggestion, we have reported error bars of the experimental results in Table 1.

- Following merB’s suggestion, we have added experiments testing the performance of SGDM and AdmetaS on finetune tasks, including SUPERB and CIFAR-10 (using deep CNN architecture) in Section G.

- Following kSFP’s suggestion, we have tried the learning rate of 0.5 of SGDM in ResNet-110 and reported the results in Section H.

We also improved other minor points of Reviewer RS9p, Reviewer dVGC, Reviewer merB, Reviewer kSFP in the revised version. Thank you all for the valuable suggestions.

Please let us know if you have additional questions.

Thank you,
Authors

---

### Decision · Program_Chairs · 2023-01-20

**Decision:**

Reject

**Justification For Why Not Higher Score:**

None of the reviewers replied when asked *why* to accept this paper, all of them gave a borderline score.

**Justification For Why Not Lower Score:**

N/A

**Metareview: Summary, Strengths And Weaknesses:**

The paper presents an variant of the Adam algorithm, replacing the exponential moving average of the gradients with a double exponential moving average and using elements from the  Lookahead optimizer. Convergence results as well as an empirical validation are presented.

The reviewers found this paper mildly interesting and promising. However, they also pointed out some shortcomings. In particular, as in many similar papers, the theoretical analysis is so bad to be almost a negative point of the paper. For example, 1) the convergence results do not show any advantage over SGD; 2) the theory does not offer any insights in the working of the algorithm; 3) the iterates are assumed to be bounded instead of proving it; 4) the analysis treats all the modifications of the algorithms as "disturbances" to SGD. Moreover, the proofs techniques are very standard. While some reviewers always tend to ask for theory, such low quality results are not really useful in any sense.
This means that the empirical comparison is the only way we have to validate the claim of the gains over the baseline algorithms.

The empirical results show some small gain, but it is very difficult to understand how significant they are. In particular, it seems extremely difficult to distinguish these gains from all the ones in all the other similar papers (accepted and rejected) from the past few years that did not survive to the test of time. Also, there are a number of hyperparameters that are not tuned for each algorithm in the experiments that might completely change the results, for example, learning rate schedule, weight decay, momentum terms, etc.

Overall, given the tepid reception from the reviewers (even after the changes done to the paper by the authors), the very weak theory, and the unclear significance of the results, I believe this paper is not ready to be published at ICLR in its current form.

**Summary Of Ac-Reviewer Meeting:**

No meeting: Reviewers didn't answer to my message about feedback on the paper.